# Differential contributions of the two cerebral hemispheres to temporal and spectral speech feedback control

Mareike Floegel [1], Susanne Fuchs [2] & Christian A. Kell [1]✉

Proper speech production requires auditory speech feedback control. Models of speech production associate this function with the right cerebral hemisphere while the left hemisphere is proposed to host speech motor programs. However, previous studies have investigated only spectral perturbations of the auditory speech feedback. Since auditory perception is known to be lateralized, with right-lateralized analysis of spectral features and left-lateralized processing of temporal features, it is unclear whether the observed right-lateralization of auditory speech feedback processing reflects a preference for speech feedback control or for spectral processing in general. Here we use a behavioral speech adaptation experiment with dichotically presented altered auditory feedback and an analogous fMRI experiment with binaurally presented altered feedback to confirm a right hemisphere preference for spectral feedback control and to reveal a left hemisphere preference for temporal feedback control during speaking. These results indicate that auditory feedback control involves both hemispheres with differential contributions along the spectro-temporal axis.

[1] Cognitive Neuroscience Group, Brain Imaging Center and Department of Neurology, Goethe University, Schleusenweg 2-16, 60528 Frankfurt, Germany. [2] Leibniz-Centre General Linguistics (ZAS), Schuetzenstr. 18, 10117 Berlin, Germany. ✉email: c.kell@em.uni-frankfurt.de

I n human verbal communication, the aim of speaking is to create an auditory percept that can easily be decoded by a listener's brain. Speech production models propose that this is achieved in two ways[1–4]. Auditory, but also somatosensory feedback of an utterance is analyzed to detect and correct mismatches between produced and intended speech. However, speech production cannot rely entirely on such slow mechanisms, because natural speech is faster than feedback control could explain[5]. Feedforward control based on internal representations of auditory-motor associations allows speaking rates that correspond to the observed speed of human communication. Nevertheless, auditory feedback contains valuable information to preserve stable and intelligible articulation in a variety of settings[6] and is used to acquire and maintain auditory-motor associations, sometimes called speech motor programs. Perturbations of the auditory speech feedback typically induce changes in articulation that compensate for the disturbance. If perturbations are experienced repeatedly and in a predictable manner, the new action-perception association is learned and speech motor programs are updated[7,8].

It is believed that feedforward and feedback control are specialized functions of the two cerebral hemispheres. The highly influential DIVA speech production model proposes that the left hemisphere is specialized in feedforward specifications of motor outputs while the right hemisphere processes auditory speech feedback to refine motor output based on external sensory information[1]. While numerous imaging studies on auditory feedback control report, indeed, right-lateralization of auditory feedback processing[9–14], other imaging studies propose speaking-related auditory-motor processing in the left hemisphere[2,5,15,16]. To date, there is no consensus on the specific contributions of the two cerebral hemispheres to auditory speech feedback control.

One important limitation of past imaging studies is the fact that only auditory speech feedback control based on spectral vowel features like fundamental frequency or formant structure has been investigated[7,17,18]. Temporal perturbations that prolong or compress speech locally change the length of phonemes and their transitions. Speech feedback control based on these temporal speech features during speaking, however, has only been described on a behavioral level[8,19]. In auditory perception, spectral processing has been linked with right-lateralized computations in the auditory cortex while the left hemisphere is thought to preferentially process temporal features of auditory stimuli[20,21]. Sensory processing is an essential part of feedback control. In consequence, it is unclear whether the observed right-lateralization in previous studies results from the exclusive use of a spectral feedback perturbation or whether it reflects right-lateralization of auditory speech feedback control in general[1], independent of the type of acoustic perturbation.

We investigated whether feedback control based on spectral and temporal speech features during speech production follows the proposed spectral/temporal distinction in functional lateralization of auditory perception. This would predict that both hemispheres contribute to auditory speech feedback control with a preference for auditory-motor processing of spectral speech features in the right and for auditory-motor processing of temporal speech features in the left hemisphere.

Healthy participants read out loud monosyllabic pseudowords while hearing their own voice (altered or unaltered in real time) via headphones. By using both vowels and consonants as perturbed speech items, we excluded the possibility that the observed lateralization results from potential hemispheric differences in processing vowels and consonants[22]. Spectral perturbations changed characteristic vowel or consonant frequencies in the acoustic domain during the entire duration of the phoneme. Vowels' (/ɪ/) first formant (F1) or consonants' (/ʃ/) center of gravity (COG) were shifted upwards by up to 20% of production. Temporal perturbations changed the length of vowels or consonants in the acoustic domain[23] and increased phoneme duration by about 50 ms. In a first speech production experiment, feedback perturbations were introduced stepwise over 40 trials (ramp phase) before the amount of perturbation was kept constant at 20% relative to production (hold phase) for another 20 trials and finally removed (after effect phase). We presented altered auditory feedback in a dichotic manner, meaning that one ear was stimulated with altered auditory feedback while the other ear simultaneously received the original, unaltered input. Dichotic stimulation biases processing in the auditory cortex to the input of the contralateral ear[24]. We predicted that compensation in response to the perturbed feedback would be greater and the produced speech output would be closer to the auditory target if altered auditory feedback was presented to the ear contralateral to the hemisphere which preferentially analyzes the perturbed speech feature. We thus expected a left ear (right hemisphere) advantage for compensating spectral auditory feedback alterations and a right ear (left hemisphere) advantage for compensating temporal feedback alterations.

While dichotic stimulation can answer questions on hemispheric specialization, it represents an unnatural experimental condition. To investigate functional lateralization of speech feedback control when identical auditory feedback is perceived with both ears, as during natural speaking, we further investigated speaking while listening binaurally to altered or normal auditory feedback during fMRI. In addition, speech adaptation was investigated by contrasting resting-state functional connectivity after and before learning new auditory-motor associations. We hypothesized that regional activity and functional connectivity profiles indicate a right hemispheric preference for spectral feedback control and a left hemisphere preference for temporal feedback control of speech.

Our data demonstrate the hypothesized functional lateralization and identify lateralized feedback control-related activations in frontal and temporal cortices. Following temporal adaptations, auditory-motor learning increases fronto-temporal interactions in the left hemisphere while spectral adaptations increase fronto-temporal resting-state connectivity in the right hemisphere.

## Results

Absolute values of produced spectral and temporal speech features in consonants and vowels were rendered comparable by normalizing them to averaged preperturbation values during baseline (see Methods). Participants expectedly changed articulation to compensate for the perturbations. We first checked whether perturbations efficiently induced compensation by investigating production changes in the binaural condition. Compared to preperturbation values, participants lowered F1 of the vowel or the COG of the fricative in response to spectral feedback perturbations (estimate $= -0.027$, SE $= 0.012$, $t(35) = -2.22$, $p = 0.03$) and shortened the vowel/fricative in response to temporal feedback perturbations increasing phoneme durations (estimate $= -0.029$, SE $= 0.012$, $t(35) = -2.37$, $p = 0.024$) with considerable interindividual variability (Fig. 1a). This effect was also observed if data of the spectral and temporal group were analyzed in two separate models (spectral: estimate $= -0.023$, SE $= 0.011$, $t(16) = -2.03$, $p = 0.05$; temporal: estimate $= -0.033$, SE $= 0.013$, $t(16) = -2.47$, $p = 0.026$). Further, production changes in response to binaurally presented perturbations were greater compared to production changes in the control condition with binaurally presented unaltered feedback ($F(3, 31)_{PertvsControl} = 6.18$, $p = 0.019$) with subthreshold $t$ values for the separate estimates (estimate$_{spectral} = 0.018$,

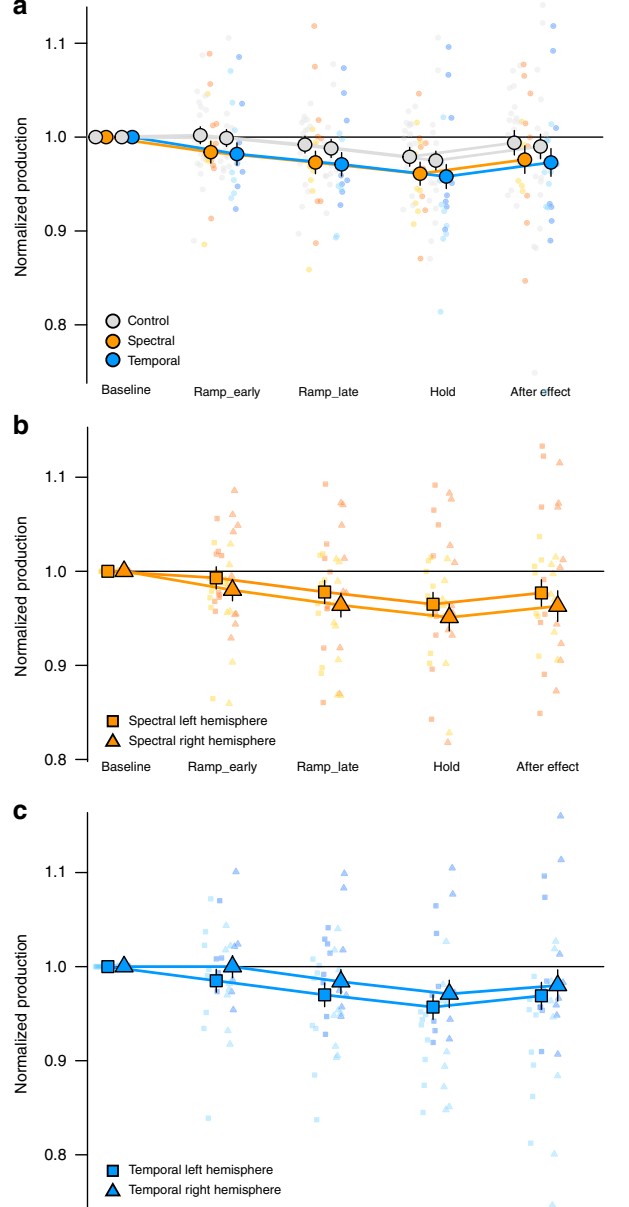

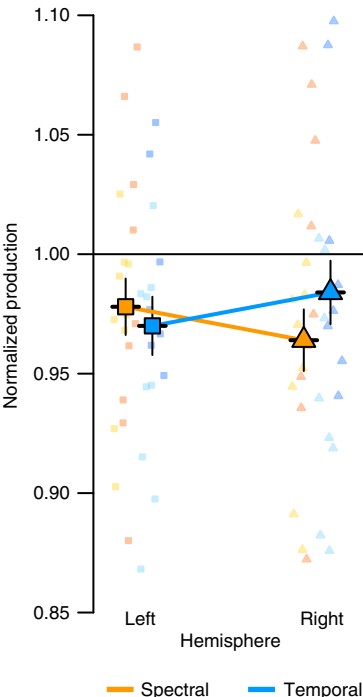

**Fig. 2 Interaction between dichotic condition (left/right) and type of feedback alteration (spectral/temporal).** Values represent produced speech features averaged across early and late ramp, hold and after effect phase separately for spectral (orange) and temporal (blue) feedback alterations presented to the left (square) and right (triangle) hemisphere, respectively. Produced speech features are relative to baseline, smaller values indicate greater compensation. Data present estimated means± model derived standard errors that account for between-subject variability ($n_{spectral} = 18$ participants, $n_{temporal} = 18$). Individual raw values are plotted in the background (yellow: spectral perturbation vowel $n = 9$, orange: spectral perturbation consonant $n = 9$, light blue: temporal perturbation vowel $n = 9$, dark blue: temporal perturbation fricative $n = 9$). Source data are provided as a Source Data file.

**Fig. 1 Compensation in the course of the dichotic listening experiment.** Main effect of block for speaking **a** with unaltered auditory feedback (gray) or speaking with binaurally presented spectral (orange) or temporal (blue) auditory feedback perturbations in blocks of twenty trials. Main effect of block for speaking with dichotically presented spectral (**b**, orange) and temporal (**c**, blue) feedback perturbations (squares correspond to left hemisphere responses, triangles correspond to right hemisphere responses). In dichotic conditions, one ear perceived the spectrally or temporally altered feedback, while the other ear perceived the unaltered, normal feedback. Produced speech features are relative to baseline, smaller values indicate greater compensation. Data present estimated means ± model derived standard errors that account for between-subject variability ($n_{spectral} = 18$ participants, $n_{temporal} = 18$ participants). Individual raw values are plotted in the background. Their colors denote participants' group allocation (yellow: spectral perturbation $n = 9$ vowel, orange: spectral perturbation consonant $n = 9$, light blue: temporal perturbation vowel $n = 9$, dark blue: temporal perturbation fricative $n = 9$). Source data are provided as a Source Data file.

SE $= 0.01$, $t(31) = 1.75$, $p = 0.09$ and estimate$_{temporal} = 0.017$, SE $= 0.01$, $t(31) = 1.63$, $p = 0.11$ Fig. 1a). Degree of compensation was largely independent of the target phoneme of the perturbation ($F(1, 31)_{VowelvsConsonantXPertvsControl} = 0.006$, $p = 0.97$; $F(1, 31)_{VowelvsConsonantXSpectralvsTemporalXPertvsControl} = 0.175$, $p = 0.68$). Expectedly, the amount of compensation did not differ between spectral or temporal perturbations when altered auditory feedback was presented to both ears ($F(1, 31)_{SpectralvsTemporalXPertvsControl} = 0.003$, $p = 0.96$).

Participants also changed their speech production to compensate for the perturbations in response to dichotically presented altered auditory feedback, the main conditions of interest in this experiment ($F(3, 34)_{Block} = 6.144$, $p = 0.002$, Fig. 1b, c). Importantly, the degree of compensation in the two dichotic conditions showed an interaction between ear and type of feedback perturbation ($F(1, 31)_{SpectralvsTemporalXLeftvsRight} = 8.47$, $p = 0.007$, Figs. 1 and 2), indicating different hemispheric preferences for auditory-motor processing of spectral and temporal speech features. Planned comparisons revealed that compensation of spectral feedback perturbations displayed a left ear/right hemisphere advantage (estimate $= 0.0139$, SE $= 0.008$, $t(38) = 1.72$, $p = 0.046$, orange in Fig. 2). In contrast, compensation of temporal feedback perturbations was greater when the right ear/

left hemisphere was presented with the prolonged phoneme (estimate = 0.0141, SE = 0.008, $t(39) = 1.71$, $p = 0.047$, blue in Fig. 2). There was a marginal trend that compensation was overall larger for dichotically presented perturbations applied to vowel compared to consonant acoustics ($F(1, 31)_{VowelvsConsonant} = 3.18$, $p = 0.08$, $estimate_{vowel} = 0.39$, SE = 0.012, $t(36) = 3.33$, $p = 0.002$; $estimate_{fricative} = 0.013$, SE = 0.012, $t(36) = 1.03$, $p = 0.31$). In isolation, this finding could potentially indicate that vowel rather than consonant perturbations induced compensatory responses in the dichotic conditions. However, the target phoneme of the perturbation (vowel or consonant) did not significantly influence the lateralization effect ($F(1, 31)_{VowelvsConsonantXLeftvsRight} = 1.41$, $p = 0.244$, $F(1, 31)_{VowelvsConsonantXLeftvsRightXSpectralvsTemporal} = 2.35$, $p = 0.14$).

This behavioral experiment employing dichotic auditory feedback perturbation indicates differential hemispheric preferences for auditory-motor processing of spectral and temporal features during speaking. However, these findings do not reveal brain regions that contribute to lateralized processing during speaking with binaural auditory feedback. We performed a sparse sampling fMRI study with 22 participants who were exposed to the identical, yet binaural, spectral auditory feedback perturbations and 22 other participants who experienced the same binaural temporal auditory feedback perturbations as in the behavioral study to answer this question.

Participants in the fMRI study also changed their speech production upon perturbation in relation to preperturbation values (Fig. 3a) with marginal significant compensation for spectral (estimate = 0.0112 SE = 0.006, $t(50) = 1.7$, $p = 0.09$) and significant compensation for temporal perturbations (estimate = 0.03 SE = 0.006, $t(50) = 4.63$, $p < 0.001$). When comparing speaking with altered auditory feedback to the control condition instead of preperturbation values, compensation was only significant for the temporal perturbation group ($F(1, 42)_{SpectralvsTemporalXPertvsControl} = 22.02$, $p < 0.001$; $estimate_{spectral} = 0.003$ SE = 0.005, $t(42) = 0.52$, $p = 0.6$; $estimate_{temporal} = 0.036$ SE = 0.005, $t(42) = 7.11$, $p < 0.001$). This resulted from carry over effects from the experimental to the control condition in the spectral perturbation group (see Fig. 3a). Whether spectral or temporal perturbations targeted the vowel or consonant did not significantly influence the amount of compensation ($F(1, 43)_{VowelvsFricativ} = 0.007$, $p = 0.93$). Participants significantly changed their speech production in response to perturbations of vowel ($estimate_{vowel} = 0.02$, SE = 0.01, $t(42) = 1.99$, $p = 0.05$) and fricative acoustics ($estimate_{fricative} = 0.021$, SE = 0.005, $t(42) = 4.03$, $p < 0.001$).

A conjunction analysis of speaking with binaurally altered auditory feedback of the vowel and consonant compared to normal speaking in the same run revealed activity associated with auditory-motor processing of spectral or temporal speech features (Fig. 3b and Table 1). Based on prior imaging studies on auditory feedback control, the search space for this analysis was restricted to auditory, ventral premotor and inferior frontal regions[25] in both hemispheres (see methods for details). Spectral control (orange in Fig. 3b) was associated with increased activity in a cluster of voxels in the right inferior frontal gyrus (IFG) that extended into the frontal operculum and in clusters in bilateral secondary auditory cortex (superior and middle temporal gyrus (STG and MTG), and superior temporal sulcus (STS)). Processing of temporal speech features during auditory feedback control (blue in Fig. 3b) was associated with increased activity in left IFG, left STG and STS, and right MTG.

To test whether activity associated with spectral and temporal auditory-motor control was indeed lateralized, we calculated weighted bootstrapped lateralization indices (LI) in the pre-defined auditory and frontal ROIs. This represents a threshold-free estimate of lateralization that considers the

extent and size of lateralization[26]. Speaking with spectrally altered auditory feedback compared to normal speaking showed strongly right-lateralized activity in frontal (LI = 0.46) and in auditory cortices (LI = 0.51, Fig. 3c). Speaking with temporally altered auditory feedback compared with unperturbed speaking, on the other hand, was strongly left lateralized in frontal (LI = −0.835) and weakly left lateralized in auditory cortex (LI = −0.116, Fig. 3c). When testing lateralization voxel-wise by flipping contrast maps and subtracting these from the original images[27], left-lateralized activity was observed in anterior portions of auditory cortex and the pars orbitalis of the inferior frontal gyrus when compensating for temporal perturbations. In contrast, spectral feedback control was associated with right-lateralized activity in anterior and posterior auditory cortex and pars triangularis of the inferior frontal gyrus (Table 2, Fig. 3d).

We further examined whether individual task-related activity in spectral and temporal feedback control areas (6 mm spheres centered on functional peak activations reported in Table 1) was related with the individual degree of compensation to spectral and temporal feedback perturbations, averaged over vowel and consonant perturbations. There was a consistent trend (illustrated in Supplementary Fig. 1) that stronger activity in left temporal feedback control areas was associated with greater compensation of temporal feedback perturbations in anterior STS ($r = 0.403$, $p = 0.063$), IFG pars triangularis ($r = 0.41$, $p = 0.059$) and IFG pars orbitalis ($r = 0.317$, $p = 0.151$). Temporal feedback control activity in right anterior STS was not related with the amount of compensation to temporal feedback perturbations ($r = 0.096$, $p = 0.669$). Similar associations between spectral feedback control activity and acoustic measures during spectral feedback perturbations were non-significant ($r_{RIFGtri} = 0.14$, $p = 0.53$; $r_{RpSTS} = 0.051$, $p = 0.821$; $r_{LpSTS} = 0.097$, $p = 0.6674$), probably due to overall lower variability in acoustic measures during spectral perturbations ($SD_{spectral} = 0.02$, $SD_{temporal} = 0.046$).

We assessed whether adapting to auditory feedback perturbations lead to learning-related changes in resting-state functional connectivity by contrasting resting-state fMRI data after and before the speech adaptation run. Learning-related plasticity should depend on the degree of compensation. Therefore, we tested whether functional resting-state connectivity between feedback control-related seeds (6 mm spheres centered on functional peak activations reported in Table 1) and the ipsilateral rest of the brain was modulated by individual production changes. Indeed, the more participants adapted speech production to spectral perturbations, the more they increased fronto-temporal connectivity in the right and temporo-parietal coupling in the left hemisphere (orange in Fig. 4, Table 3). In contrast, acquiring new auditory-motor associations following temporal perturbations of the auditory feedback was associated with increased fronto-temporal coupling only in the left hemisphere. In addition to the increased coupling with the left inferior frontal gyrus, the left auditory association cortex connected more strongly with the left supplementary motor area (SMA) following temporal perturbations (blue in Fig. 4, Table 3).

## Discussion

Both the behavioral and the neuroimaging study provide evidence for bihemispheric, yet asymmetric, auditory feedback control of speaking. Right-lateralization of auditory feedback control[9–14] resulted from spectral perturbations. Temporal speech features, in contrast, were more strongly processed by the left hemisphere. Learning-related neuroplasticity increased fronto-temporal interactions in the right hemisphere following spectral perturbations and in the left hemisphere following temporal perturbations.

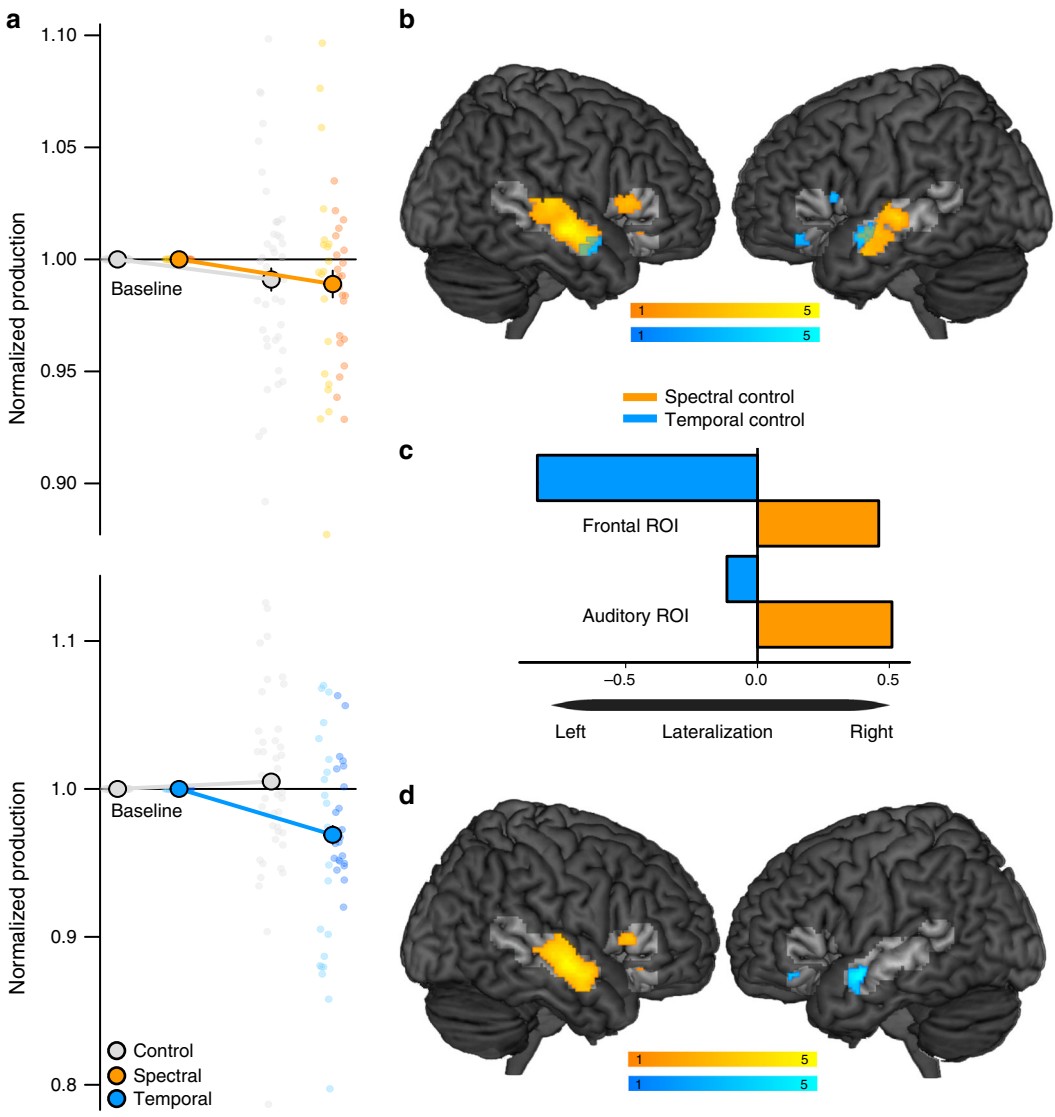

**Fig. 3 Speech production with binaurally presented spectral and temporal feedback perturbations during fMRI. a** Behavioral data. Produced speech feature relative to baseline during speaking with normal (gray) or spectrally (orange) or temporally (blue) altered auditory feedback. Smaller values indicate greater compensation. Data present estimated means ± model derived standard errors that account for between-subject variability ($n = 22$ participants per group). Individual raw values for the control condition (grey) and following perturbations of the vowel (light colors $n = 22$ participants per group) or consonant (dark colors $n = 22$ participants per group) are plotted in the background. **b** Brain areas that activate during speaking with spectrally (orange) or temporally (blue) altered auditory feedback compared to normal speaking (perturbation$_{vowel}$ > no perturbation ∩ perturbation$_{consonant}$ > no perturbation), at $p < 0.05$ FWE, small volume cluster corrected for multiple comparisons with a cluster forming threshold of $p < 0.001$ (one-sided, paired $t$-test, $n = 22$ participants per group, color represents $t$-values, see color scales, search space is highlighted). Overlap in activity between spectral and temporal feedback control is shown in green. **c** Weighted lateralization indices in frontal and auditory ROIs during spectral (orange) and temporal (blue) feedback control. Negative values indicate left-lateralization, positive values right-lateralization. **d** Lateralized brain areas during speaking with spectrally (orange) or temporally (blue) altered auditory feedback compared to normal speaking (Lateralization$_{pertvowel>noperturbation}$ ∩ Lateralization$_{pertfric>noperturbation}$) at $p < 0.05$ FWE, small volume cluster corrected for multiple comparisons with a cluster forming threshold of $p < 0.001$ (one-sided, paired $t$-test, $n = 22$ participants per group, color represents $t$-values, see color scales, search space is highlighted). $P$-values for panels **b** and **d** are provided in Tables 1 and 2. Source data of panel **a** and **c** are provided as a Source Data file.

Our findings suggest a modification of prevailing speech production models. Previous theoretical models propose a single auditory feedback controller, either in the right hemisphere[1] or did not specify the contributions of the two cerebral hemispheres[3]. Previous studies have already demonstrated involvement of both hemispheres during speaking[9,14,28–30]. We specify here that fronto-temporal cortices in both hemispheres serve speech feedback control. We propose a refined theoretical speech

production model based on the DIVA model, because it constitutes the only neurocomputational model that takes contributions of both hemispheres into account[1].

The DIVA model proposes a feedback control system in the bilateral superior temporal cortex that compares external auditory feedback with sensory predictions from a left-lateralized feed-forward system in the frontal lobe. A potential mismatch signal is transferred to the right IFG, which is thought to constitute the

**Table 1 Clusters of activation during speaking with spectrally (columns two to five) or temporally altered feedback (columns six to nine) compared to normal speaking (conjunction over consonant and vowel perturbations), small volume corrected for multiple comparison in literature-based ROIs at $p < 0.05$ with a cluster defining threshold of $p < 0.001$ (one-sided, paired $t$-test, $n = 22$ participants).**

| Region | $k_{spec}$ | $p_{spec}$ | $T_{spec}$ | Coordinates x/y/z | $k_{temp}$ | $p_{temp}$ | $T_{temp}$ | Coordinates x/y/z |
|---|---|---|---|---|---|---|---|---|
| R STG/STS | 769 | <.001 | 5.86 | 60/−14/−10 | 49 | .03 | 3.05 | 56/2/−20 |
| L STG/STS | 330 | <.001 | 4.17 | −56/−20/−2 | 82 | .015 | 3.38 | −48/−8/−8 |
| R IFG (tri) | 38 | .027 | 3.18 | 50/22/6 | | | | |
| L IFG (tri) | | | | | 44 | .025 | 3.08 | −42/20/10 |
| L IFG (orb) | | | | | 47 | .023 | 2.71 | −38/38/−12 |

*k* cluster size, *R* right, *L* left, *STG* superior temporal gyrus, *STS* superior temporal sulcus, *IFG* inferior frontal gyrus, *tri* pars triangularis, *orb* pars orbitalis.

**Table 2 Lateralized brain activity during speaking with spectrally (columns two to five) or temporally (columns six to nine) altered feedback compared to normal speaking (conjunction over consonant and vowel perturbations), small volume corrected for multiple comparison in literature-based ROIs at $p < 0.05$ with a cluster defining threshold of $p < 0.001$ (one-sided, paired $t$-test, $n = 22$ participants per group).**

| Region | $k_{spec}$ | $p_{spec}$ | $T_{spec}$ | Coordinates x/y/z | $k_{temp}$ | $p_{temp}$ | $T_{spec}$ | Coordinates x/y/z |
|---|---|---|---|---|---|---|---|---|
| R STG/STS | 647 | <.001 | 4.78 | 60/−12/−8 | | | | |
| L STG/STS | | | | | 67 | .026 | 3.82 | −48/0/−14 |
| R IFG (tri) | 17 | .05 | 2.63 | 50/22/6 | | | | |
| L IFG (orb) | | | | | 16 | .05 | 2.33 | −36/38/−12 |

*k* cluster size, *R* right, *L* left, *STG* superior temporal gyrus, *STS* superior temporal sulcus, *IFG* inferior frontal gyrus, *tri* pars triangularis, *orb* pars orbitalis.

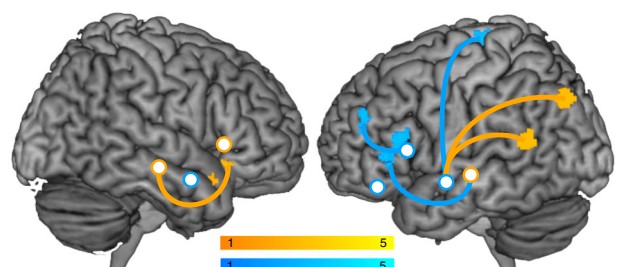

**Fig. 4 Resting-state functional connectivity in response to spectral and temporal adaptation.** Intrahemispheric resting-state functional connectivity changes associated with the learning of new auditory-motor associations following spectral (orange) and temporal feedback perturbations (blue) (perturbation$_{vowel}$ ∩ perturbation$_{consonant}$), cluster level corrected for multiple comparisons at $p < 0.05$ FWE with a cluster forming threshold of $p < 0.001$ (one-sided, paired $t$-test, $n = 22$ participants per group, color represents $t$-values, see color scales). Seed regions (white circles) are based on functional activation maxima during speaking with spectrally (orange) or temporally (blue) altered feedback compared to normal speaking (see Fig. 3b and Table 1). Note that the right inferior frontal cluster extends from the lateral surface into the orbitofrontal cortex, which seemingly gives rise to two clusters. Spectral adaptation also increased connectivity between L aSTS and L IPS and between R pSTS and HG (Table 3, these deeper structures are not rendered on the surface).

only region that translates deviations from sensory predictions into corrective motor commands that in turn are fed into the left-lateralized feedforward system. Our data indicate that the right auditory association cortex monitors preferentially spectral speech features while the left auditory association cortex preferentially monitors temporal speech features (see Fig. 5). This

functional lateralization bases upon a hemispheric specialization in auditory processing[20,21]. Mismatches are forwarded to the respective ventral inferior frontal gyri which amplifies functional lateralization in the frontal cortices. Our data suggest that in the left hemisphere, the ventral IFG adapts primarily temporal properties of motor commands, which results in updated motor timing or adapted velocities of articulator actions. In the right hemisphere, the ventral IFG likely corrects the articulatory targets of the motor command, which affects position of the articulators and in turn results in spectral adaptations of the acoustic speech signal. This specifies the external auditory feedback control system in such a way that it consists of two parallel loops in the two hemispheres. Functional lateralization does not indicate a dichotomy but rather a slight shift in the equilibrium between functional homologues. It is thus likely that information in both hemispheres is integrated on multiple levels via interhemispheric interactions[29,31].

Because lateralization studies on different aspects of somato-sensory feedback processing are lacking, we did not specify the contributions of the two cerebral hemispheres to somatosensory feedback control in our model. The few imaging studies on somatosensory feedback processing during speaking suggest a comparable functional lateralization as in the auditory domain. Somatosensory feedback processing during articulation is associated with left-lateralized activity in the supramarginal gyri[16,32]. Perturbations of somatosensory feedback increase activity in the bilateral supramarginal gyri and in the right ventral IFG[33], possibly because the studied perturbation required adapting position of articulators more than their velocities.

The main new feature of the external auditory feedback loop, the parallel processing of auditory information in both hemispheres, can also be incorporated into external feedback loops of state feedback control models of speech production. However, in contrast to the DIVA model, those models propose an additional

**Table 3 Resting-state functional connectivity between seeds and the whole brain associated with spectral (columns three to six) and temporal (columns seven to ten) sensorimotor adaptation (conjunction over consonant and vowel adaptation), cluster corrected for multiple comparisons at $p < 0.05$ with a cluster defining threshold of $p < 0.001$ (one-sided, paired $t$-test, $n = 22$ participants per group).**

| Seed | Region | $k_{spec}$ | $p_{spec}$ | $T_{spec}$ | Coordinates x/y/z | $k_{temp}$ | $p_{temp}$ | $T_{spec}$ | Coordinates x/y/z |
|------|--------|------|------|------|------|------|------|------|------|
| R pSTS | R HG | 59 | .051 | 3.00 | 48/−12/0 | | | | |
| | R IFG (orb) | 47 | .022 | 2.45 | 38/22/−10 | | | | |
| L aSTS | L IPS[a] | 115 | <.001 | 3.74 | −32/−52/34 | | | | |
| | L IPL | 98 | <.001 | 2.85 | −28/−76/38 | | | | |
| | L TPJ | 118 | <.001 | 2.97 | −54/−52/20 | | | | |
| | L SMA | | | | | 57 | .006 | 3.27 | −8/−26/47 |
| L pSTS | L IFG (tri) | | | | | 99 | <.001 | 2.63 | −50/28/18 |
| L IFG (tri) | L MFG | | | | | 55 | <.001 | 2.51 | −30/44/20 |

k, cluster size; R, right; L, left; a, anterior; p, posterior; HG, Heschl's gyrus; IFG, inferior frontal gyrus; orb, pars orbitalis; IPS, intraparietal sulcus; IPL, inferior parietal lobule; TPJ, temporo parietal junction; SMA, supplementary motor area; STS, superior temporal sulcus; tri, pars triangularis; MFG, middle frontal gyrus.
[a]Not illustrated.

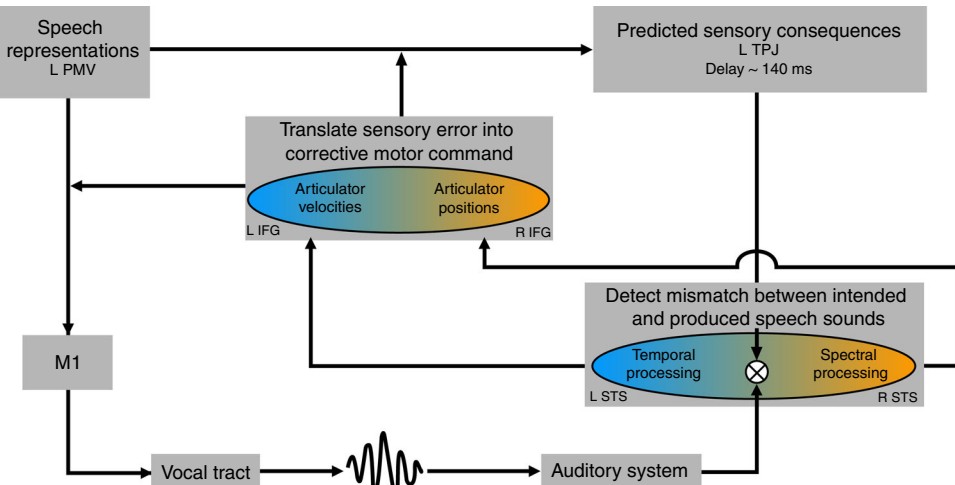

**Fig. 5 Revised auditory feedback control loop of the DIVA model.** The specified model includes two parallel auditory feedback control subsystems that preferentially monitor and correct temporal (blue) or spectral (orange) properties of the produced speech sound. Projections from the left premotor cortex to the bilateral auditory association cortices transmit the expected sensory consequences of the produced speech sound via the left TPJ, both for normal speaking and for speaking with altered auditory feedback[4]. The predicted auditory consequences of speaking are compared with the incoming auditory information taking the temporal delay of feedback information into account. The left STG monitors especially temporal and the right STG especially spectral speech features. Potential mismatches in either domain are transmitted via temporo-frontal projections and translated into corrective commands for the timing of articulator movements in left IFG and corrective commands that specify articulator target positions in right IFG. Corrective motor commands are transmitted to the feedforward system targeting M1 and the vocal tract and to the internal model for updating auditory-motor associations. Neither potential interhemispheric connections nor somatosensory feedback control loops are shown for the sake of clarity. PMV ventral premotor cortex, IFG inferior frontal gyrus, STG superior temporal gyrus, TPJ temporo-parietal junction.

internal feedback loop that estimates sensory consequences of articulation even in the absence of overt motor behavior and external feedback[3,15]. Learning new auditory-motor associations should affect also internal representations of actions and their sensory consequences. The observed changes in resting-state functional connectivity point to plasticity associated with the updating of internal models. Updating temporal properties of auditory-motor associations increased interactions between the ventral IFG and the auditory association cortex only in the left hemisphere. In contrast, new auditory-motor associations in response to spectral perturbations increased coupling between the right auditory association cortex and the right ventral IFG. This suggests that both hemispheres update their auditory-motor speech representations. The observation of, both, a left and a right internal fronto-temporal loop appears in contradiction with the

proposal of a single left-lateralized internal auditory-motor interface in the left temporo-parietal junction (TPJ)[2,15]. However, our resting-state data confirmed that the left TPJ was associated with auditory-motor learning. This suggests that left TPJ plays a special role in relaying top-down predictions from the left-lateralized feedforward to the bihemispheric feedback control system. The fact that left TPJ changed its functional connectivity primarily when learning new auditory-motor associations following spectral perturbations was surprising, since a role of the right TPJ in internally representing spectral speech features could have been envisaged. However, the right TPJ did not play a role in this context. A greater involvement of left TPJ in internally representing spectral compared to temporal speech features raises the question whether temporal adaptions also involve another region linking the feedforward to the feedback system. The left

auditory association cortex connected more strongly with the left SMA following temporal perturbations. The SMA has been implicated in action timing and in processing temporal aspects of sensory input[34–38]. We thus propose the SMA as an additional motor-to-auditory interface that internally translates the temporal structure of articulator actions into expected auditory consequences. In Fig. 5, this would translate into two parallel inputs into the feedback control system, one preferentially predicting temporal speech features via the left SMA and one preferentially predicting spectral speech features via left TPJ. Because this proposal needs to be backed up by additional empirical evidence we did not yet implement this into our model.

Why should the brain process spectral and temporal aspects of auditory input differently in the two hemispheres? Parallel processing of complex sensory information allows for rapid and efficient responses but asks for separating processing chains to a certain degree. Such separation has been proposed to result from differential filtering of the input in sensory association cortices[20,21,39,40]. A right hemisphere preference for spectrally tuned and a left hemisphere preference for temporally tuned auditory receptive fields has been reported earlier[41,42] but was only recently shown for speech stimuli[21,43]. Differential sensitivity of the two cerebral hemispheres to temporal and spectral modulation rates in acoustic signals has been proposed to result in auditory representations with high temporal and low spectral resolution in the left hemisphere compared to auditory representations with low temporal and high spectral resolution in the right. Accordingly, also the right auditory association cortex represents temporal information, although with relative low temporal resolution, during speech perception[44] or during non-speech auditory-motor control[38]. In our experiments, the temporal perturbation increased phoneme duration by about 50 ms. Controlling such temporal changes requires high temporal resolution and thus a sensitivity for high temporal modulation rates. Detecting spectral changes of about 20% relative to production, on the other hand, requires analyzing spectral modulation rates with sufficient resolution. Consequently, auditory-motor control based on spectral speech features lateralizes to the right. Temporal speech features in the studied range represent phonemic contrasts, such as short versus long vowels, single versus geminate consonants, or voiced versus voiceless consonants[45] and thus code linguistic information. Also, temporal stretching and shrinking of segments mark prosodic boundaries[46] and stress, as well as accent[45]. Spectral speech features represent linguistic information concerning vowel identity, sonorants, and place of articulation of consonants[47]. In addition, spectral features code speaker's gender, age, size, dominance and turn taking behavior[48–50].

We observed considerable interindividual variability in the degree of compensation that was only partially explained by regional brain activity or functional connectivity. As in other speech perturbation studies[7,51], individuals differ in the degree and sometimes even direction of compensation and more work is needed to understand the sources of such interindividual variability.

One could argue that the observed lateralization in speaking-related brain activity associated with spectral and temporal perturbations represents a trivial consequence of bottom-up processing of speech with different temporal and spectral characteristics[21]. However, functional asymmetries during speaking were not only observed in auditory but also in inferior frontal regions that have been associated with compensation of perturbed auditory feedback[9,25]. More importantly, the differential involvement of the left and right hemisphere in temporal and spectral auditory feedback control also manifested in traces of auditory-motor learning in the resting-state data. We thus interpret the observed functional lateralization during speaking as evidence of active auditory-motor

processing serving speech control in both sensory and nonsensory cortices. In contrast, connectivity between cerebral speech production regions and the cerebellum[52] was not increased by auditory-motor learning.

Of note, feedback control is only one component of producing fluent and intelligible speech. Other functions during speech production like e.g., speech forward control, syntactic or semantic processing, may follow other principles of hemispheric specialization[53]. A far greater role of the left compared to the right hemisphere in feedforward control of speech has been well established[54]. In the context of auditory speech feedback control, however, the right hemisphere seems to be as important as the left hemisphere to improve articulation by means of sensory information.

Our results indicate that both hemispheres are involved in the processing of auditory speech feedback to control articulation, contrary to the view of a general right hemisphere preference for feedback control during speech production. We identified one factor that determines the degree to which both hemispheres contribute to feedback control of speaking. The present study highlights that compensating and learning new spectral auditory-motor associations recruits primarily the right hemisphere while compensating and learning new temporal auditory-motor associations recruits especially the left hemisphere during speech production.

## Methods

**Participants.** Forty healthy volunteers (20 female) participated in the behavioral experiment and 44 healthy volunteers (27 female) in the fMRI study. Participants were adult right-handed native speakers of German (handedness score[55] behavioral study mean = 93, SD = 9.8, fMRI study mean = 90, SD = 11.5) and reported normal speech and hearing. All participants gave their written informed consent before participation. Four participants had to be excluded from the behavioral experiment because real-time tracking of formants ($n = 1$) and vowel and fricative portions ($n = 2$) did not consistently work or task instructions have not been followed ($n = 1$, singing instead of speaking). The study was approved by the ethics committee of the Medical Faculty of Goethe-University Frankfurt (DFGKE 1514/2-1) and was in accordance with the Declaration of Helsinki.

**Behavioral experiment.** Sixteen experimental manipulations were studied in a mixed within and between subject design to reduce the number of conditions per participant, which is important because parallel implicit learning of new auditory-motor associations has so far only been reported for up to three different perturbations[56]. Participants were evenly divided across four experimental groups that differed with respect to the acoustic property that was altered throughout the experiment. Participants either experienced spectral or temporal perturbations of the vowel or the consonant in their auditory speech feedback. Each participant experienced four different conditions (binaural unaltered feedback, binaural altered feedback and two dichotic conditions).

Participants read words out loud, speaking into a microphone (AT 2010, Audio-Technica) placed 10 cm in front of them and were told that they heard their utterances via headphones (DT770 PRO, beyerdynamics). The level of auditory feedback provided by the headphones was amplified (~+15 dB relative to the level at the microphone) to reduce the influence of unaltered bone conducted auditory feedback[7,18,19].

Altered auditory feedback was either presented to both ears (binaurally), to only the right ear or to only the left ear while the other ear received the unaltered auditory feedback (dichotic conditions). In the latter two conditions, auditory processing is biased to the input of the contralateral ear[24]. To ensure that changes in speech production were related to auditory speech feedback perturbations and not just a side-effect of word repetition, the original, unaltered feedback was presented to both ears in an additional control condition.

To enable the acquisition of several distinct auditory-motor transformations in parallel, each of the four feedback presentation modes was associated with a different, predictable word context[56]. The real-time feedback alteration targeted always the same part of monosyllabic CVC-pseudowords (either the vowel or second consonant) while the other phonemes implicitly distinguished between conditions. Within participants, the allocation of word context to feedback condition was consistent. Over participants, the association between word context and feedback condition was counterbalanced.

Experimental stimuli were chosen such that they (1) facilitated online consonant and vowel tracking, (2) could be altered in the spectral and temporal domain, (3) provided context information to learn multiple auditory-motor transformations in parallel, and (4) introduced as little acoustic variability between

syllables as possible. To facilitate the algorithmic distinction (which relied on the presence/absence of voicing) between vowels and consonants in CVC-pseudowords, the second consonant was voiceless. Further, contrary to most studies targeting the vowel /e/ for spectral feedback perturbations, we have chosen the high vowel /ɪ/ to reduce the likelihood of glottalisation[57]. This was of importance, since glottalization interferes with the temporal feedback perturbation of the vowel. Spectral and temporal perturbations of the consonant targeted always the voiceless fricative /ʃ/, which can be altered to be perceived as /s/. The preceding vowel was chosen in such a way that global shifts in the fricative spectrum due to lip rounding were kept at a minimum, i.e., it was not a rounded vowel. This was of importance, since such coarticulatory shifts in the frequency spectrum would have biased perception of /ʃ/ into the direction of /s/[58]. In consequence, CVC-pseudowords for perturbations of the vowels were [bɪʃ], [bɪɾ], [bɪç], and [bɪs]. CVC-pseudowords for perturbations of the consonants were [bɪʃ], [bɛʃ], [baʃ] and [bœʃ].

The behavioral feedback alteration experiment consisted of five ten minute blocks. In every block, each word-condition pair was presented 20 times resulting in a total of 400 trials. The experiment started with a baseline block in which auditory feedback was presented unaltered in all four conditions. Afterwards, feedback alterations were introduced gradually in steps of 5% relative to speech production over 40 trials (ramp_early/ramp_late). Auditory feedback in the control condition remained unaltered throughout the whole experiment. The perturbation was kept at a maximum (20% relative to production) for another 20 trials per feedback presentation mode (hold) until feedback was returned to normal (no alteration) in all conditions for another 20 trials per condition (after effect).

Following the adaptation task, participants were asked whether they noticed something special during speaking. Five participants noticed that sometimes auditory feedback seemed to differ between the right and the left ear. Another four participants noticed that the words sometimes sounded different. No participant identified the type of auditory speech feedback perturbation or noticed a change in the way he/she spoke.

Data analysis and statistics are reported below.

**fMRI experiment.** Twenty-two participants experienced a temporal perturbation of their auditory speech feedback, the other 22 participants were studied during a spectral perturbation of their auditory speech feedback.

The experiment started with normal speaking without auditory speech feedback perturbations to ensure that differences between resting-state measurements (see below) were not driven by prolonged exposure to scanner noise or adapting to hear the own voice via headphones (preperturbation baseline). In addition, this run's behavioral data were used as baseline values and served for normalization of behavioral data during the feedback perturbation run.

Resting-state scans (7 min each) were acquired to assess changes in functional connectivity due to the learning of new auditory-motor speech associations in the spectral or temporal domain. The resting-state scans were acquired before (preadaptation) and after speaking with altered speech feedback (postadaptation). Participants were instructed to have their eyes open during the measurement and to fixate a white cross on black background in the middle of a screen.

In between the resting-state scans participants performed the feedback perturbation run.

Participants read out loud three different, visually presented CVC-pseudowords ([bɪʃ], [dɪʃ], and [gɪʃ]) while they heard their own voice (altered or unaltered) mixed with white noise through headphones[8,17,18]. Each pseudoword was associated with one of three experimental conditions (no perturbation, vowel perturbation, consonant perturbation). This allowed to delineate feedback control processes during speech production from other speech production processes and to generalize results over vowels and consonants. The rationale for choosing these three syllables was the same as for the behavioral experiment. Yet, the same vowel and fricative was used in all syllables to reduce acoustic variability of the speech token even further. Only the initial plosive served as contextual cue to learn multiple sensory-motor associations in parallel[56]. Participants' speech was recorded with an MR-compatible microphone (FOMRI-III™ noise cancelling microphone, Optoacoustics) and fed back via OptoActive™ active noise cancelling headphones (Optoacoustics). The level of auditory feedback provided by the headphones was amplified to reduce the influence of unaltered bone conducted auditory feedback resulting in 90 dB headphone output.

In contrast to the behavioral study, auditory feedback perturbations were kept constant at 20% relative to production throughout the whole fMRI run[59,60]. The three CVC-pseudowords were presented in randomized order, 30 times each. The presentation of syllables was interspersed (one quarter of trials) with the presentation of a nonspeech condition where participants should remain silent, saw the letter string "yyyy", and heard white noise[9,25].

To allow participants to speak in relative silence and to reduce movement artifacts we used an event-related sparse sampling technique[9,25]. Each trial started with the 2 s long acquisition of one functional image. Image acquisition was followed by a pause of 0.5–1.5 s after which the CVC pseudoword or the nonspeech stimulus was visually presented for 2 s. After another pause of 2.5–3.5 s, the next image was acquired resulting in a total trial length of 8 s. The jittered acquisition delay accounted for variability in the timing of the BOLD response depending on participant and brain region[61]. The jitter was chosen in such a way to sample BOLD responses around their estimated peaks 4–7 s after speech onset.

Participants were instructed to speak with normal conversational loudness and duration. In case participants did not speak loud enough to detect vowel onsets in their utterance, a prompt to speak louder was displayed. Participants were naïve to the fact that their auditory feedback would be altered throughout the experiment. Before the actual start of the experiment (but already inside the scanner), participants were trained to get familiarized with the experimental setup, particularly the way their own (unaltered) voice sounds via headphones.

**Data acquisition.** Microphone input and headphone output was digitally sampled at 48 kHz and recorded at 16 kHz.

Scanning was performed using a Siemens (Erlangen, Germany) Trio 3 Tesla magnetic resonance scanner with a commercial eight-channel coil. High-resolution T1-weighted anatomical scans (TR = 1.9 s; TE = 3.04 ms; flip angle = 9°; 192 slices per slab; 1 mm³ isotropic voxel size) were obtained to improve spatial normalization of functional images onto the Montreal Neurological Institute (MNI) brain template. Functional images were obtained with a gradient-echo T2*-weighted transverse echoplanar image (EPI) sequence (Task-fMRI (122 volumes; TR = 2 s; TE = 30 ms; silent gap = 6 s; flip angle = 90°; 32 axial slices; 3 mm³ isotropic voxel size), Resting-State (178 volumes; TR = 2 s; TE = 30 ms; flip angle = 90°; 30 axial slices; 3 mm³ isotropic voxel size)).

**Auditory feedback perturbations.** All real-time tracking and perturbing was performed using the Matlab Mex-based digital signal processing software package Audapter[23].

To enable presentation of altered and unaltered auditory feedback in parallel in the two dichotic conditions, an additional temporary buffer was introduced into Audapter to duplicate the incoming audio signal before it was downsampled and altered. While a duplicate of the incoming signal was processed to introduce a feedback alteration, the original signal was held in a temporary buffer and transferred unaltered to the output. To compensate for the delay of the signal processing algorithm, the unaltered, original signal was delayed the same amount as the altered signal. The altered output was transferred to one output channel while the unaltered but also delayed signal was transferred to the other one.

The online status tracking function of Audapter was used to restrict feedback perturbations to either the vowel or the fricative in the syllable. Vowel onsets were tracked by an empirically defined root-mean-squared intensity threshold. Fricative onset was defined as the time point when the ratio of spectral intensity in high vs. low frequency bands crossed an empirically defined threshold for more than 0.02 s.

The vowel /ɪ/ was perturbed spectrally by increasing its F1 up to 20% relative to production (fixed formant perturbation method in Audapter). F1 is perceptually relevant to distinguish vowel sounds from each other and correlates positively with tongue height and mouth openness during articulation[47]. Perceptually, the spectral vowel alteration of /ɪ/ resulted in an acoustic signal closer to the vowel /e/. The formant shift procedure introduced a temporal 11 ms delay. The consonant /ʃ/ was perturbed spectrally by increasing its spectral centroid (amplitude-weighted mean frequency of a speech spectrum) in the acoustic output up to 20% relative to production. The spectral centroid is an important characteristic to distinguish the two fricatives /s/ and /ʃ/ at a perceptual level and correlates with the place of articulation[47], i.e., it is higher for the alveolar place of articulation than the postalveolar fricative. Thus, the spectral alteration of the /ʃ/-sound resulted in an acoustical signal whose spectral centroid was closer to the fricative /s/. The Audapter algorithm for changing the spectral centroid shifts the whole-frequency spectrum and thus also its amplitude-weighted mean. The spectral centroid alteration introduced a 24 ms delay between input and output (see Supplementary Fig. 2 for an illustration of the spectral perturbations).

Temporal perturbations either increased the length of the vowel or fricative in the acoustic output. The vowel and consonant were perturbed temporally by increasing their length up to 20% relative to production. Vowel and consonant length was increased by time warping in the frequency domain. The time warping event was configured such that time dilation spanned the whole vowel/consonant. The average vowel/consonant length was estimated based on vowel/consonant productions in the training phase. The rate of the catch-up period was set to 2 resulting in a natural sounding acoustic output. In total, the time-warp perturbation introduced a 24 ms delay between input and output (see Supplementary Fig. 2 for an illustration of the temporal perturbations).

**Behavioral data analysis.** Vowel- and consonant boundaries were marked manually according to the recording's speech waveform and a broadband spectrogram (window size 5 ms) in PRAAT[62]. Onsets and offsets of vowels and fricatives were labelled from an in-house annotator. CVC-productions of participants, who experienced a temporal feedback perturbation, were additionally labelled by an external annotator who was blinded with respect to the experimental procedure and speech feedback perturbation. This was important to ensure that changes in length estimates were not unconsciously influenced by knowledge and expectations about experimental alterations. Analyses on length estimates were therefore only performed on the data labelled by the external annotator. Inter-rater agreement with the in-house annotator was good (ICC(2, 1)$_{\text{Consonant}}$ = 0.83, [0.79–0.859]; ICC(2, 1)$_{\text{vowel}}$ = 0.869, [0.847–0.886]).

F1 estimates of the vowel, COG estimates of the consonant and relative vowel and consonant lengths for each CVC-production were extracted in PRAAT. The average F1 value of an utterance was calculated in a time window of 40–80% of the vowel duration[63] using the burg algorithm. The maximum frequency of formants for female speakers was set to 5500 Hz and for male speakers to 5000 Hz. The spectral centroid was calculated in a time window of 40–80% of the fricative duration. The signal was high-pass filtered at 1000 Hz before spectral centroid calculation[64]. The spectral centroid was calculated as the weighted mean of a frequency spectrum obtained by a Fast Fourier transform. Before spectral centroid calculation, the spectrum was cepstrally smoothed with 500 Hz to reduce the influence of spectral outliers on spectral centroid estimates[65]. To assess vowel and fricative length changes, we calculated relative segment lengths to account for different speaking rates between trials. The relative vowel length was calculated by subtracting vowel onset from consonant onset and dividing this duration by the whole-word length. The relative consonant length was calculated by dividing the consonant length by word duration. All trials in which the perturbed speech parameter deviated more than ±2 standard deviations from its mean within a block were discarded. F1, COG and relative length estimates were rendered comparable across alterations, stimuli and speaker by a normalization procedure that divided each produced speech feature with its average production during preperturbation baseline. This results in comparable values of relative production changes that were used for statistics, while the raw values are in different units (Hz and ms).

Linear mixed effects models (LMM) were used to test whether participants changed their produced speech features in response to spectral and temporal feedback perturbations. We modelled binaural and dichotic conditions separately. The first LMM on binaural data served to check whether the spectral and temporal perturbations induced compensatory responses. Specifically, we tested whether participants changed articulation over the course of the experiment, whether the type and/or the target phoneme of the feedback perturbation modulated compensation, and whether compensation was greater compared to a control condition with normal auditory speech feedback. To this end we entered block (ramp early/ramp late/hold/after effect phase), feedback alteration (altered/unaltered), type (spectral/temporal), and target of the feedback alteration (vowel/consonant) as fixed effects into the model and allowed by-subject random slopes for the effect of block and feedback alteration. Due to the normalization procedure, compensation in relation to preperturbation values was assessed by comparing marginal estimated model means of perturbed productions against 1, separately for the spectral and temporal groups, using two-sided, paired t-tests. We additionally assured that the effects were equally observed when data were modelled separately for the spectral and temporal groups (two separate models with identical factors, see above).

The data from the dichotic conditions were investigated in another LMM that tested whether the produced speech features in response to dichotically presented spectral and temporal auditory feedback perturbations depended on which ear received the perturbed auditory feedback, the central research question in this experiment. In this model, block (ramp early/ramp late/hold/after effect phase), ear (left/right), type (spectral/temporal), and target of the auditory feedback alteration (vowel/consonant) were entered as fixed effects. We allowed by-subject random slopes for the effect of ear and block. The binaural control condition was not entered into this model because functional lateralization was assessed by contrasting data from the dichotic conditions directly with each other. Significant type × ear interactions were followed up by planned comparisons averaging overall blocks following the baseline. Planned comparisons tested whether production changes in response to spectral or temporal feedback alteration differed significantly between the two dichotic conditions with spectral alterations showing a left ear/right hemisphere advantage and temporal alterations a right ear/left hemisphere advantage (one-sided, paired t-tests on marginal estimated means).

The behavioral data of the fMRI study were analyzed with an LMM analogous to the first LMM in the behavioral experiment and allowed testing whether produced speech features with altered auditory feedback were significantly different from baseline productions and a control condition in the same experimental run. The model contained type of feedback alteration (spectral/temporal), feedback alteration (altered/unaltered), and the target of the feedback alteration (vowel/consonant) as fixed effects and allowed by-subject random slopes for the feedback alteration and target of the feedback alteration. Comparisons with baseline productions were again investigated by testing whether marginal estimated means for production changes in response to spectral and temporal perturbations differed significantly from 1 (two-sided, paired t-test).

P-values were provided by the Satterthwaite's degrees of freedom method. Linear mixed effects models, planned comparisons and post-hocs on estimated marginal means were performed with the *afex*[66] package (version 0.20-2) in R version 3.5.3.

**Imaging data analysis**. Image processing and statistical analysis was performed using SPM 12[67]. All results were visualized using MRIcron[68]. Imaging data are available at https://identifiers.org/neurovault.collection:7569.

The spatial preprocessing pipeline used standard SPM 12 parameters complemented by additional steps to account for possible motion due to speaking. The pipeline encompassed the following steps: (1) Realignment of functional images using rigid body transformation, (2) coregistration of subject's individual structural scans with the mean functional image of the realignment step, (3) smoothing of images with an isotropic 4 mm full-width at half-maximum Gaussian

kernel to prepare images for additional motion adjustment with Art Repair[69], (4) motion adjustment of functional images with ArtRepair to reduce interpolation errors from the realignment step, (5) normalization of functional images to a symmetric brain template via parameters from segmentation of structural scans, and (6) another smoothing of images with an isotropic 7 mm full-width at half-maximum Gaussian kernel. The symmetric brain template was created by averaging the standard Montreal Neurological Institute (MNI) brain template within the Talairach and Tournoux reference frame with its R/L flipped version[29]. The preprocessed functional images were analyzed within the framework of general linear models (GLM) adapted for nonspherical distributed error terms.

The GLM contained three regressors of interest, modelling the three auditory feedback conditions (no perturbation, vowel perturbation, consonant perturbation). Due to the additional motion adjustment step during preprocessing movement-related effects were not modelled additionally[69]. Condition-specific regressors were obtained by convoluting the onset and duration of conditions (modelled by boxcar functions) with the canonical hemodynamic response function. To account for the use of a sparse sampling protocol, we adjusted microtime resolution and onset (SPM.T = 64, SPM.$T_0$ = 8 s). The model was high-pass filtered with a cutoff at 128 s to remove low frequency drifts. An autoregressive model AR(1) was used to account for serial autocorrelations in the time series.

After model estimation, two contrasts were specified testing the effect of speaking with altered auditory feedback (vowel or consonant perturbation) against normal speaking without perturbation in each individual (first-level). The resulting contrast images were subjected to second level random effect analysis to infer brain activation at the population level. Data from the spectral and temporal perturbation groups were analyzed separately in two repeated measure ANOVAs. To exclude the possibility that any feedback control-related effects were due to differences in the processing of vowels or consonants we only investigated effects that were consistent across feedback perturbations of the vowel and consonant. Thus, we investigated conjunctions across both contrasts testing the global null hypothesis[70] to identify spectral/temporal feedback control regions. The global null hypothesis reveals all those brain areas that are consistently activated throughout both conditions and jointly significant. Given the common practice to increase statistical power in sparse sampling fMRI experiments via region-of-interest (ROI) analyses[9,14,71], we investigated activity differences between conditions within a small volume restricted search space that spanned literature-based ROIs for auditory feedback control. An auditory and a frontal ROI was defined a priori based on 10 mm spheres centered on previously reported functional activation maxima for speaking with altered auditory feedback compared to normal speaking in a random effects whole brain analysis[25] (Supplementary Table 1 and illustrated as lighter cortex in Fig. 3). Because this study focuses on the contribution of both hemispheres to feedback control, we included homotopic regions of the reported activation maxima into the ROIs. Family-wise error correction was performed at $p < 0.05$ at the cluster level with a cluster defining threshold of $p < 0.001$ small volume corrected in aforementioned auditory and frontal ROIs. Activation coordinates are given in MNI space.

Lateralization was first assessed using weighted bootstrapped lateralization indices (LI) with the LI toolbox in SPM[26]. LIs were calculated on the SPMs representing activity associated with spectral and temporal feedback control in the auditory and ventral frontal ROI. Weighted LIs with a negative sign indicate left-lateralized activity while weighted LIs with a positive sign indicate right-lateralized activity. While weighted bootstrapped LIs provide a robust and threshold-free method to assess lateralization[26] they lack spatial sensitivity. We thus also calculated voxel-wise laterality maps by flipping feedback-related first-level contrast images (speaking with altered feedback > normal speaking) along the interhemispheric fissure and subtracting these flipped mirror images from the original (unflipped) contrast images. The obtained spectral and temporal laterality maps were subjected to two additional repeated measurement ANOVAs. Again, we tested the conjunction over vowel and consonant contrasts to asses which voxels showed higher activity in one hemisphere compared to the other during spectral or temporal feedback control. Family-wise error correction was performed at $p < 0.05$ at the cluster level with a cluster defining threshold of $p < 0.001$, small volume corrected in the auditory and frontal ROIs. Activation coordinates are given in MNI space.

The relationship between participants' individual degree of compensation to spectral and temporal feedback perturbations and individual activity in spectral and temporal feedback control areas was assessed using Pearson's correlations. In analogy to the aforementioned fMRI analyses we did not dissociate vowel and consonant effects and correlated averaged vowel and consonant productions with averaged activity during vowel and consonant perturbations. Similar to the fMRI compensation contrast, degree of compensation was defined as the difference between speech features during perturbation and speech features during the control condition in the same run. Feedback control regions were defined post-hoc according to functional peak activations in the contrast speaking with spectrally or temporally altered auditory feedback compared to normal speaking. They consisted of 6 mm spheres centered on local peak activation maxima for spectral (bilateral posterior STS and right IFG triangularis) or temporal (bilateral anterior STS, left IFG triangularis and orbitalis) feedback control (Fig. 3b Table 1). Correlations were tested at $p < 0.05$, uncorrected for multiple comparisons.

Functional connectivity at rest was analyzed with the Conn toolbox[72]. Images were spatially preprocessed with the same preprocessing pipeline described above.

In addition, time-series were denoised to reduce the impact of physiological noise and motion on results. Physiological noise was removed with the anatomical component-based noise correction method (aCompCor) and 16 orthogonal time-courses in subject-specific WM and CSF ROIs[72]. Further, subject-specific motion parameters and their first derivative (scan-to-scan motion), task-effects and subject-specific time points identified as outliers (scan-to-scan global signal change >9 and movement more than 2 mm) were regressed out. To isolate low frequency fluctuations, resting-state data were bandpass filtered (0.008–0.09 Hz)[72].

For each participant and each resting-state run (pre- and postadaptation) seed-to-voxel connectivity maps were generated by calculating bivariate correlations between the average seed time-series and the whole brain. Seeds for the connectivity analysis were the same 6 mm spheres centered on local peak activations of the spectral and temporal feedback control contrast that served for correlation analyses with degree of compensation. The second level GLM contained two regressors representing changes in connectivity between resting-state runs (one for the spectral group and another for the temporal group) and four parametric regressors that represented the subject-specific amount of compensation for spectral or temporal perturbations of the vowel and consonant, separately. The parametric regressors were included to identify connectivity changes between resting-state runs that were associated with motor learning of the new spectral and temporal auditory-motor associations due to feedback control. Connectivity changes between resting-state runs at the average level of compensation were captured by the first two regressors and not analyzed further. With this model we assessed motor learning-related connectivity changes by means of conjunction analyses over vowel and consonant regressors (e.g., post/preadaptation difference that correlates with F1 compensation ∩ post/preadaptation difference that correlates with COG compensation). SPMs were thresholded at $p < 0.05$ FWE corrected at the cluster level with a cluster defining threshold of $p < 0.001$.

**Reporting summary**. Further information on research design is available in the Nature Research Reporting Summary linked to this article.

## Data availability
The unthresholded statistical parametric maps that support the findings of this study have been deposited at https://neurovault.org with the access code https://identifiers.org/neurovault.collection:7569. The source data underlying Figs. 1, 2, 3a, 3c and supplementary Fig. 1 are provided as a Source Data file. Source data are provided with this paper.

## Code availability
All analyses were performed using Matlab R2012b and R version 3.5.2, with standard functions and toolboxes (see Methods). All code is available upon request. Source data are provided with this paper.

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

## Acknowledgements
We thank Olivia Maky for technical help. This study was funded by the German Research Foundation with an Emmy Noether Grant to CAK (KE 1514/2-1).

## Author contributions
M.F., C.A.K., and S.F. designed the experiments. M.F. collected and analyzed the data. S.F. analyzed data. M.F. and C.A.K. prepared the manuscript. S.F. edited the manuscript. C.A.K. supervised the project and acquired funding.

## Competing interests
The authors declare no competing interests.
