## [Peer Review File · Nature Communications]

Reviewers' Comments:

Reviewer #1:

Remarks to the Author:

Floegel and colleagues study feedback control of uttered speech to answer the question whether each hemisphere is involved differently in the analysis of spectral vs temporal analysis in this process. They hypothesise that the right hemisphere preferentially analyses spectral speech properties and therefore provides feedback on spectral perturbations, while the left hemisphere preferentially analyses temporal speech properties. Their data suggest that this is the case.

This is overall an excellent study with a well thought-through paradigm and analyses. It adds significantly to the literature. I particularly like use of two different behavioural and two different neuroimaging analyses. The study is well embedded in, and motivated by, the speech perception context, although it is noteworthy that the spatial/temporal distinction in perception is not as clear cut as presented here (some studies find a right-lateralisation in temporal envelope tracking). The manuscript is also well written and was a joy to read. I have a few minor/medium comments that should be considered to further improve the manuscript.

Minor/medium comments

1. Please also show the behavioural data of the fMRI experiment.
2. After looking at the supplementary information, it seems that the Frontal ROI consists of two ROIs (ventral premotor and inferior frontal regions). This is a bit unclear in the text. It is also counterintuitive to analyse these regions jointly, especially as they seem to show differential effects in Fig. 2, and seem to be separate seeds for the connectivity analysis. In my opinion, it would make sense to also analyse these separately.
3. In addition to showing a lateralisation by subtracting activity of one hemisphere, it would also make sense to add a direct statistical comparison of activity between hemispheres in the ROIs. Please report this as well.
4. Please show plots that allow to assess the distribution of values in Figure 1a,b,c. Means and standard errors are not sufficient to assess the data.
5. The results section can only be fully appreciated once the reader has read the methods section. For example, it is unclear what "auditory-motor learning/associations" refers to or what "Baseline - Ramp_early - etc." in Figure 1 means. It's also hard to interpret the results without knowing what was analysed at this point. Please add a few sentences to the results section to explain what was done and what the stimuli looked like.
6. In Figure 1a, does the black dotted line reflect compensation to both temporal and spectral binaural perturbations? It is said in the text that there is no difference between these conditions, but it is hard to believe that both results are identical at each time point, so shouldn't there be two lines for the binaural condition?
7. On page 6 it says "Whether spectral or temporal perturbations were applied on vowel or consonant acoustics did not significantly influence the degree of compensation in the binaural [...] or in the dichotic conditions". The statistics however, indicate a marginal effect that should be mentioned. In contrast, a p-value of 0.11 is positively interpreted as "subthreshold" (page 10, line 189), where it fits the narrative. Such a double dissociation should be avoided.
8. It would be good to spell out somewhere what the compensation looked like behaviourally. I.e. participants shortened/lowered their utterances when they heard the increased phoneme durations/pitch.
9. In the revised model, it would be good to show which elements belong to the original DIVA model and which to the revised one. Maybe use different shades of grey for old/new elements?
10. It is difficult to reconcile the regions in Figure 4 with the regions in Figure 3. It would for example help to add abbreviations for seed regions and connected regions to Fig 3. Alternatively, a brain plot

could be added to the model in Fig 4 to illustrate the regions and processing flow.

11. The manuscript should cite this directly relevant work: Flinker, A., Doyle, W. K., Mehta, A. D., Devinsky, O., & Poeppel, D. (2019). Spectrotemporal modulation provides a unifying framework for auditory cortical asymmetries. *Nature human behaviour*, 3(4), 393.

12. How were seed regions defined for the connectivity analysis?

13. For the manual marking of on-/offsets of vowels/fricatives, which measure was used for analysis (rater 1, rater 2, average)?

14. "COG" is not defined.

Signed: Dr Anne Keitel, University of Dundee, UK

Reviewer #2:

Remarks to the Author:

This study provides behavioral, fMRI task-based, and fMRI resting-state data pertaining to the question of how feedback during speech production is handled in the two cerebral hemispheres. The study has a number of strengths. The hypothesis that is explored is based on a solid and highly cited model of speech production, but the authors introduce an important modification to the feedback component of this model. Whereas most instantiations of it suggest that feedback is handled primarily by a right-hemisphere mechanism, the authors propose that different kinds of feedback may be handled differentially within each hemisphere. This idea arises from models of spectro-temporal functional differences between left and right hemispheres, which have until now been separate from the speech production models. By integrating these two theoretical ideas, they achieve an interesting synthesis that moves the field forward. The other strength is the converging findings from the different experimental manipulations, all of which point in the same direction, thus lending more credence to the overall argument.

There are a few points for improvement that I would suggest

1. The presentation of the data could be enhanced. In Fig 1 for instance, b) and c) show the same data in two ways, but neither gives enough information on individual differences. I would like to see some representation of the distribution, such as by showing each individual data point superimposed upon the bars. Similarly for the behavioral data of the fMRI experiment, all we are told is that there was a significant effect, but there is no way to know what the mean values and distributions were like; they should at the very least be stated, if not illustrated. Were the adaptation effects similar for the two experiments? Fig 1a is a little hard to follow with all the different lines; would it be possible/valuable to show the values expressed as difference from baseline? It might make it easier to see the main effects.

2. Some aspects of the fMRI data shown in Fig 2 were unclear to me. It appears this image represents a contrast analysis of after feedback to before feedback. But then on p. 9 the authors indicate the importance of relating such effects more directly to individual behavior, which I certainly agree with. Yet, instead of demonstrating that the effects in Fig 2 are correlated with behavioral indices of spectral or temporal modulation, they perform a less straightforward analysis in which behavior is correlated with hemispheric differences. I am not sure I understand the rationale for this analysis, which anyway is not presented very clearly, since only a single r-value is given for each hemisphere (even though there is more than one ROI), and since no scatterplot is provided to be able to inspect the data for outliers, nonlinear trends, and so forth.

Fig 2a look like some kind of voxelwise analysis, but on line 155 it is stated that ROI analyses were conducted, even though Fig 2a is referred to. If Fig 2a is an ROI analysis, as indicated by the text,

then why do the clusters look different in each hemisphere in terms of size and shape? The caption seems to indicate a totally different analysis, involving some kind of conjunction of vowel and consonant data, but this is not described in the text, which is confusing. Also, the ROIs mentioned are auditory, IFG, and ventral premotor, but then the results (line 159) indicate something in the insula, which is not only invisible in the figure, but also not one of the ROIs.

3. I could not figure out the relationship between the data shown in Fig 3 and Table 2. Unless I am mistaken they are supposed to show the same analysis (as indicated on line 202), but they don't seem to correspond. For instance, in Table 2, the left anterior STG shows modulation with three different left-hemisphere areas during the spectral manipulation; so there should be three orange lines emanating from this region in the upper right panel of Fig 3; but there are only two such lines. The left posterior STG is indicated as having an increased connectivity with the left TPJ, also in the spectral condition, so there should be an orange line for that, but it's not present. Conversely, in the figure there are two orange lines for the right posterior STG, but this latter region does not exist in the table. There are other discrepancies too.

More generally I was not very convinced of the value of the interhemispheric connectivity findings. I am not sure what hypothesis this analysis was supposed to test, and the discussion barely mentions these findings. I would question whether it is meaningful to include these findings without a better framing of the intention behind them, or of their interpretation.

Minor items:

Although perhaps not essential, it might be nice for the reader to have an illustration of the stimuli used and the nature of the spectral or temporal manipulation applied. This could be done with spectrograms of the stimuli before and after the modification was made for example.

I found this wording (line 225) very odd: "Our findings ask for a specification of prevailing speech production models..." Perhaps the authors mean to say that their findings suggest a modification, or an addition to existing models.

"principals" should be "principles" on line 344

Reviewer #3:

Remarks to the Author:

This manuscript investigates the claim that during speaking, auditory feedback control in the spectral domain (e.g. compensating for mismatches in formants, pitch, or spectral centroid) is differentially processed by the right hemisphere, while control in the temporal domain (e.g. compensating for mismatches in phoneme or syllable timing) is differentially processed by the left hemisphere. This is a straightforward claim that makes clear predictions beyond those made by existing models of speech production, which do not delineate separate contributions of temporal or spectral feedback processing circuits. These predictions are tested in two well-designed complementary experiments and are borne out by the authors' data. First, adaptation to altered auditory feedback is differentially enhanced or diminished based on the ear that is receiving the feedback alteration: there is more adaptation when hearing a spectral mismatch in the left ear (right hemisphere) or a temporal mismatch in the right ear (left hemisphere). Second, the authors present here the first neuroimaging study of temporal perturbations to speech feedback, showing that responses are left-lateralized in the ROIs that are responsive to these perturbations, while responses to spectral perturbations are right-lateralized. These are novel results that touch on both sensorimotor control of speaking and hemispheric specialization for auditory stimuli and would be of wide interest to the speech and language

community.

The claims would be made more convincing by showing the data, especially the behavioral adaptation, in more detail. In Figure 1, the dichotic listening conditions are collapsed across two groups each (i.e. across vowel and consonant perturbations), and the binaural condition (black dotted line) seems to be averaging across all four groups (i.e., across both spectral and temporal perturbations). While seeing this aggregate data paints a clear picture of the overall effect, I think it's more appropriate to show spectral and temporal groups separately. These are very different manipulations (an increase in produced F1 vs. a decrease in spectral center of gravity vs. a shortening of phoneme duration) and the raw compensation measurements are on different scales (Hz vs. ms). Even if the compensation evoked by these different types of feedback perturbations is normalized, and is comparable in magnitude once normalized, it's still an overreach to average together these different data types and not show the data from the individual conditions. Since participants either experienced spectral or temporal perturbations, it would make sense to split these data and show all of the four conditions (binaural, left, right, and control) on each plot, at least in a supplemental figure if not in the main text. (Further, in the current Figure 1, the error bars for the five lines are difficult to discern; it would help to offset them horizontally or include error bar "caps" so it is easier to tell where the error bars for each line begin and end.)

The adaptation data from the fMRI study are not shown at all, so there is no way to evaluate the magnitude of the behavioral effects in this study. In particular, since different statistical reliabilities are reported for the spectral and temporal groups ($p = 0.05$ vs. $p < 0.001$), a behavioral data figure would be useful to evaluate the compensation each type of perturbation evoked. These separate statistics for spectral and temporal perturbations are not given for the behavioral study; it is unclear why they are separated here and not in the behavioral study, since in both studies there were separate groups of participants for the two types of feedback perturbations.

Relatedly, since neither behavioral data nor correlation plots are included for the fMRI study, there is no sense of range for the adaptation effects and how they contribute to the observed neural activation. Furthermore, showing a correlation between adaptation and whole-brain activity (perturbation > no perturbation), rather than the correlation between adaptation and cross-hemisphere differences, would be a more primary measure and useful for drawing conclusions about how different regions are recruited in the course of auditory-motor learning.

Methods:

- Throughout the paper, "perturbations" and "manipulations" seem to be used interchangeably. This is somewhat confusing especially given the abstract where they might be interpreted contrastively (the abstract twice contrasts "temporal manipulations" with "spectral perturbations").

- Were the 40 participants in the behavioral experiment evenly divided among the four experimental groups (10 per group)?

- Line 400: "We have chosen the high vowel /i/..." and line 515: "The vowel /i/ was perturbed..." -- Do you mean the vowel /ɪ/? According to lines 409 and 444, spoken stimuli do not contain the vowel /i/.

- How were the written stimuli spelled when presented to participants?

- The total fMRI trial length was reported to be 8 seconds, but the trial timeline seems to have events that sum to 9 seconds (2s acquisition + 0.5-1.5s pause + 2s CVC presentation + 3.5-4.5s pause).

- Was perceived pitch also altered by Audapter during the spectral perturbation of /f/? Did participants compensate for this by lowering their F0?

Results:

- Any overlap of regions responding to both spectral and temporal perturbations should be denoted in Figure 2a (it is difficult to tell if there is no overlap or if activation from one group of subjects is merely on top of the other).

Discussion:

- Lines 281-283: The SFC model doesn't propose a single left-lateralized internal auditory-motor interface in the left TPJ. While Hickok et al. do claim Spt as the primary sensorimotor integration area, Houde and Nagarajan's model does not commit to this laterality: "Note that although, for simplicity, only the neural substrate in the left hemisphere is shown here, we would expect the full network of the neural substrate to include analogous areas in the right hemisphere as well. At this point, the SFC model is agnostic regarding hemispheric dominance in the proposed neural substrate."

- Line 337-338: "increased executive control" is a somewhat speculative interpretation based only on increased coupling between auditory association cortices and fronto-parietal control networks.

Responses to Reviewer 1

Floegel and colleagues study feedback control of uttered speech to answer the question whether each hemisphere is involved differently in the analysis of spectral vs temporal analysis in this process. They hypothesise that the right hemisphere preferentially analyses spectral speech properties and therefore provides feedback on spectral perturbations, while the left hemisphere preferentially analyses temporal speech properties. Their data suggest that this is the case.

This is overall an excellent study with a well thought-through paradigm and analyses. It adds significantly to the literature. I particularly like use of two different behavioural and two different neuroimaging analyses. The study is well embedded in, and motivated by, the speech perception context, although it is noteworthy that the spatial/temporal distinction in perception is not as clear cut as presented here (some studies find a right-lateralisation in temporal envelope tracking). The manuscript is also well written and was a joy to read. I have a few minor/medium comments that should be considered to further improve the manuscript.

We thank all reviewers for their positive evaluation and their very constructive remarks. We now discuss also Keitel et al., Plos Biol 2018 for speech-envelope tracking in the lower frequency range. We appreciate the contribution of the right hemisphere to tracking all lower frequency bands which we believe requires only relatively low temporal resolution.

1. Please also show the behavioural data of the fMRI experiment.

Done, please see Figure 3a.

2. After looking at the supplementary information, it seems that the Frontal ROI consists of two ROIs (ventral premotor and inferior frontal regions). This is a bit unclear in the text. It is also counterintuitive to analyse these regions jointly, especially as they seem to show differential effects in Fig. 2, and seem to be separate seeds for the connectivity analysis. In my opinion, it would make sense to also analyse these separately.

We believe we have not stated clearly in which way the ROIs were used for analyses in the previous version of our manuscript. The ROIs were used as search space for voxel-wise analyses (i.e. small volume correction), which preserves spatial specificity (please see also our response to reviewer 2, comment 2b). The correlation and functional connectivity analyses are based on significant activations in the search space, because we were not interested in correlations and connectivity of the entire search space. We now clearly label those set of voxels as “spheres” for correlation and “seeds” for connectivity analysis. Please note that this approach does not represent double-dipping (Kriegeskorte et al 2009. Nature Neuroscience, 12, p 535–540). The only analysis in which data from the entire frontal and temporal ROIs were extracted is the weighted lateralization index analysis which requires activation-independent ROI definitions. Because the lateralization index is comparable in the two frontal ROIs, there is no additional information for the reader. We thus chose to stick to our previous version of the Figure. Please find the requested information below.

Weighted lateralization indices in the auditory, inferior frontal and premotor ROI during spectral (orange) and temporal (blue) feedback control. Negative values indicate left-lateralization, positive values right-lateralization

3. In addition to showing a lateralisation by subtracting activity of one hemisphere, it would also make sense to add a direct statistical comparison of activity between hemispheres in the ROIs. Please report this as well.

Thanks for this suggestion. Please note that we assessed lateralization not simply by subtracting activity in one hemisphere from the other, but by calculating weighted bootstrapped LIs (Wilke & Schmithorst 2006. *NeuroImage*, 33 (2), pp. 522-530). Weighted LIs do not represent the activity difference across hemispheres, but differences in activated voxels across hemispheres divided by the total amount of activated voxels. Weighted LIs are threshold-free by calculating LIs across several thresholds and computing the weighted average of these. Further, the implemented approach is quite robust to outliers due to repeated resampling and the calculation of a trimmed mean LI. We added a more detailed description of the bootstrapped LI in the methods section (p. 28 section Lateralization).

Weighted LIs provide information on functional lateralization of rather large search volumes and thus do not have spatial resolution within the ROIs. We added a voxel-wise lateralization analysis in the ROIs. This analysis revealed significantly lateralized clusters of activity in the right inferior frontal gyrus, pars triangularis and pars orbitalis for spectral feedback control and in the left inferior frontal gyrus, pars orbitalis for temporal feedback control. The results are reported in Table 1 and illustrated in Figure 3c.

4. Please show plots that allow to assess the distribution of values in Figure 1a,b,c. Means and standard errors are not sufficient to assess the data.

We apologize for this omission. We now additionally plot individual values that are color coded and reveal group allocation (see also our response to reviewer 3, comment 1). The lines in the binaural conditions are no longer aggregated across spectral and temporal perturbations (see also comment 6 and response to reviewer 3 comment 1). For sake of clarity, we now present two separate figures.

5. The results section can only be fully appreciated once the reader has read the methods section. For example, it is unclear what “auditory-motor learning/associations” refers to or what “Baseline – Ramp_early – etc.” in Figure 1 means. It’s also hard to interpret the

results without knowing what was analysed at this point. Please add a few sentences to the results section to explain what was done and what the stimuli looked like.

We added an explanation of our perturbation paradigm at the end of the introduction (p. 5). We now write: *“In a first speech production experiment, feedback perturbations were introduced stepwise over 40 trials (ramp phase) before the amount of perturbation was kept constant at 20% relative to production (hold phase) for another 20 trials and finally removed (after effect phase)”*.

Further, we added a more detailed description of the analyses to the results section. We now write: *“A conjunction analysis of speaking with binaurally altered auditory feedback of the vowel and consonant (compared to normal speaking) revealed activity associated with auditory-motor processing of spectral or temporal speech features (Figure 3b and Table 1). Based on prior imaging studies on auditory feedback control, the search space for this analysis was restricted to auditory, ventral premotor and inferior frontal regions (Niziolek & Guenther 2013. The Journal of Neuroscience, 33(29), pp. 12090-12098) in both hemispheres (see methods for details).”* (p. 8). ...

“We further examined whether individual task-related activity in spectral and temporal feedback control areas (6mm spheres centered on functional peak activations reported in Table 1) was related with the individual degree of compensation to spectral and temporal feedback perturbations (Supplementary Figure 1)” (p. 9). ...

“We assessed whether adapting to auditory feedback perturbations lead to learning-related changes in resting state functional connectivity by contrasting resting state fMRI data after and before the speech adaption run” (p. 9)

6. In Figure 1a, does the black dotted line reflect compensation to both temporal and spectral binaural perturbations? It is said in the text that there is no difference between these conditions, but it is hard to believe that both results are identical at each time point, so shouldn't there be two lines for the binaural condition?

Indeed, in our first version of the manuscript we aggregated data from the spectral and temporal group, because the difference in production in response to spectral and temporal feedback perturbations was minimal. We now show the data for both groups separately in Figure 1.

7. On page 6 it says “Whether spectral or temporal perturbations were applied on vowel or consonant acoustics did not significantly influence the degree of compensation in the binaural [...] or in the dichotic conditions”. The statistics however, indicate a marginal effect that should be mentioned. In contrast, a p-value of 0.11 is positively interpreted as “subthreshold” (page 10, line 189), where it fits the narrative. Such a double dissociation should be avoided.

Thanks for pointing this out. We now explicitly mention the trend that responses to perturbations of the vowel may be greater than responses to perturbations of the consonant. Please note that this does not influence the interaction between type of perturbation and ear advantage.

8. It would be good to spell out somewhere what the compensation looked like behaviourally. I.e. participants shortened/lowered their utterances when they heard the increased phoneme durations/pitch.

We added such a description in the beginning of the results section. It reads now: “*On average, participants lowered F1 of the vowel or the COG of the fricative in response to spectral feedback perturbations (estimate = -0.027, SE = 0.019, t(35) = -2.31, p = 0.027) or shortened the vowel/fricative in response to temporal feedback perturbations (estimate = -0.028, SE = 0.012, t(35) = -2.33, p = 0.026) in response to increased phoneme durations with considerable interindividual variability (Figure 1).*” (p. 6).

Because compensatory responses vary between participants, we added a paragraph discussing interindividual variability in compensatory responses (p. 14).

9. In the revised model, it would be good to show which elements belong to the original DIVA model and which to the revised one. Maybe use different shades of grey for old/new elements?

The new elements are illustrated in color, because we did not add any more boxes to the DIVA model, but specified lateralization within boxes. Note that in most DIVA illustrations there is no TPJ box, but a direct arrow from ventral premotor cortex to auditory state maps. However, Frank Guenther explicitly proposed a relay in left TPJ in his 2016 publication (Guenther & Hickock 2016. *Neurobiology of Language*, pp. 725-740). We therefore chose to illustrate TPJ as a box. Because this addition is not based on our data, we did not color it differently.

Further, our revised model does not make any conclusions regarding the somatosensory feedback system. We decided to add a paragraph to the discussion stating why we left out this part of the DIVA model. It reads: “*Because lateralization studies on different aspects of somatosensory feedback processing are lacking, we did not specify the contributions of the two cerebral hemispheres to somatosensory feedback control in our model. The few imaging studies on somatosensory feedback processing during speaking suggest a comparable functional lateralization as in the auditory domain. Somatosensory feedback processing during articulation is associated with left-lateralized activity in the supramarginal gyri (Agnew et al. 2013. *NeuroImage* 73, pp. 191-199; Kell et al. 2017. *Human Brain Mapping* 38, 493–508). Perturbations of somatosensory feedback increase activity in the bilateral supramarginal gyri and in the right ventral IFG (Golfinopoulos et al. 2011. *NeuroImage* 55, 1324–1338.), possibly because the studied perturbation required adapting position of articulators more than their velocities.*” (p 11.).

10. It is difficult to reconcile the regions in Figure 4 with the regions in Figure 3. It would for example help to add abbreviations for seed regions and connected regions to Fig 3. Alternatively, a brain plot could be added to the model in Fig 4 to illustrate the regions and processing flow.

We provide here a brain plot illustrating regions and processing flows. Please note that the brain plot includes interhemispheric connections between the feedforward (dark regions) and feedback components (colored regions) which have not been in the focus of this experiment and therefore are rather speculative. Actually, we believe that: “*information in both hemispheres is integrated on multiple levels via interhemispheric interactions*” (p. 11). Because these questions require further investigation, we would prefer not to include the brain plot into the manuscript. However, we would be happy to do so per request of the reviewers and editors.

Revised auditory feedback control loop of the DIVA model. Panel b shows regions and processing flows on a brain plot for the two cerebral hemispheres.

11. The manuscript should cite this directly relevant work: Flinker, A., Doyle, W. K., Mehta, A. D., Devinsky, O., & Poeppel, D. (2019). Spectrotemporal modulation provides a unifying framework for auditory cortical asymmetries. *Nature human behaviour*, 3(4), 393.

We agree with the reviewer that the work by Flinker and colleagues is highly relevant for the current study. It is cited in the introduction (p. 4) and discussion (p. 11 and 13). We also cited our very recent publication on hemispheric difference in controlling timing of finger tapping (Pflug et al. 2019, eLife.), which shows that the reported principle of lateralization is not necessarily speech-specific.

12. How were seed regions defined for the connectivity analysis?

Seed regions for the connectivity analysis were defined according to functional peak activation coordinates of the spectral and temporal feedback control contrast, because we were not interested in correlations and connectivity of the entire search space. To clarify the definition of seed regions we rephrased the description in the methods section. This part reads now: “Seeds for the connectivity analysis were the same 6mm spheres centered on local peak

activations of the spectral and temporal feedback control contrast that served for correlation analyses with degree of compensation“ (p. 30).

In addition, we added a short explanation to the results section: *“Therefore, we tested whether functional resting-state connectivity between feedback-control related seeds (6mm spheres centered on functional peak activations reported in Table 1) and the ipsilateral rest of the brain was modulated by the degree of compensation” (p. 9).*

See also our response to comment 2.

13. For the manual marking of on-/offsets of vowels/fricatives, which measure was used for analysis (rater 1, rater 2, average)?

We used the measures of the external annotator, because she was blind with regard to data to condition mapping. This is now explicitly stated on p 24.

14. “COG” is not defined.

Done.

Response to Reviewer 2

This study provides behavioral, fMRI task-based, and fMRI resting-state data pertaining to the question of how feedback during speech production is handled in the two cerebral hemispheres.

The study has a number of strengths. The hypothesis that is explored is based on a solid and highly cited model of speech production, but the authors introduce an important modification to the feedback component of this model. Whereas most instantiations of it suggest that feedback is handled primarily by a right-hemisphere mechanism, the authors propose that different kinds of feedback may be handled differentially within each hemisphere. This idea arises from models of spectro-temporal functional differences between left and right hemispheres, which have until now been separate from the speech production models. By integrating these two theoretical ideas, they achieve an interesting synthesis that moves the field forward. The other strength is the converging findings from the different experimental manipulations, all of which point in the same direction, thus lending more credence to the overall argument.

There are a few points for improvement that I would suggest

1. The presentation of the data could be enhanced. In Fig 1 for instance, b) and c) show the same data in two ways, but neither gives enough information on individual differences. I would like to see some representation of the distribution, such as by showing each individual data point superimposed upon the bars. Similarly for the behavioral data of the fMRI experiment, all we are told is that there was a significant effect, but there is no way to know what the mean values and distributions were like; they should at the very least be stated, if not illustrated. Were the adaptation effects similar for the two experiments?

We apologize for our previous omission. Per request of all reviewers, we now show all the individual behavioral data points (please see our answer to reviewer 1 comment 4 and 6 and reviewer 3). We now illustrate the results in the dichotic conditions in a single panel in Figure 2 showing individual values from both ears/hemispheres separately. Figure 3 now illustrates the behavioral data during fMRI and shows comparable changes in relation to baseline as in the behavioral study.

Fig 1a is a little hard to follow with all the different lines; would it be possible/valuable to show the values expressed as difference from baseline? It might make it easier to see the main effects.

We believe that the new Figure 1 now nicely illustrates compensation. Note that the lateralization effect is now much better visible.

2a. Some aspects of the fMRI data shown in Fig 2 were unclear to me. It appears this image represents a contrast analysis of after feedback to before feedback. But then on p. 9 the authors indicate the importance of relating such effects more directly to individual behavior, which I certainly agree with. Yet, instead of demonstrating that the effects in

Fig 2 are correlated with behavioral indices of spectral or temporal modulation, they perform a less straightforward analysis in which behavior is correlated with hemispheric differences. I am not sure I understand the rationale for this analysis, which anyway is not presented very clearly, since only a single r-value is given for each hemisphere (even though there is more than one ROI), and since no scatterplot is provided to be able to inspect the data for outliers, nonlinear trends, and so forth.

We agree with the reviewer that the previous correlation between interhemispheric balance of activity and degree of compensation was not straight forward. Per request, we removed this analysis from the manuscript and, as suggested, now correlated the individual degree of compensation with activity values. The methods and results section were revised accordingly. (p. 9 and p. 29). We report the correlations separately for each region and show scatterplots illustrating the relationship between brain activity and behavior as a supplementary Figure 1. There was a consistent trend that activity in the left hemisphere was associated with the degree of compensating temporal feedback perturbations, while a similar effect for spectral perturbations was not observed, likely due to smaller interindividual variability in the degree of compensation for spectral perturbations.

Please note that the fMRI activations represent the contrast between speaking with perturbation compared to speaking without feedback perturbation. We clarified this in the methods section.

2b. Fig 2a look like some kind of voxelwise analysis, but on line 155 it is stated that ROI analyses were conducted, even though Fig 2a is referred to. If Fig 2a is an ROI analysis, as indicated by the text, then why do the clusters looks different in each hemisphere in terms of size and shape? The caption seems to indicate a totally different analysis, involving some kind of conjunction of vowel and consonant data, but this is not described in the text, which is confusing. Also, the ROI's mentioned are auditory, IFG, and ventral premotor, but then the results (line 159) indicate something in the insula, which is not only invisible in the figure, but also not one of the ROIs.

We apologize for not being clear enough in the previous version of our manuscript. As mentioned in our response to reviewer 1, comment 2, we indeed illustrated results of a voxelwise analysis in the previous Figure 2a, now Figure 3b. We now state more clearly that the ROIs served to restrict the search space of the voxel-wise analysis to literature-based brain regions (small volume correction). We now illustrate the restricted search space for the analysis directly in Figure 3 instead of our previous Supplementary Figure 1. This standard approach did not influence the claim of our paper. For transparency, we provide here whole brain results at an uncorrected threshold ($p < 0.001$). We also upload the unthresholded statistical maps to a repository.

Brain areas that activate during speaking with spectrally (orange) or temporally (blue) altered auditory feedback compared to normal speaking ($perturbation_{vowel} > no\ perturbation \cap perturbation_{consonant} > no\ perturbation$), at $p < 0.001$ uncorrected. Overlap is displayed in green.

The voxelwise analysis, as all fMRI analyses in this manuscript, represents a conjunction analysis over vowel and consonant data. To make this clearer, we added a sentence to the results section, which reads:” *A conjunction analysis of speaking with binaurally altered auditory feedback of the vowel and consonant compared to normal speaking in the same run revealed activity associated with auditory-motor processing of spectral or temporal speech features (Fig. 3b and Table 1). Based on prior imaging studies on auditory feedback control the search space for this analysis was restricted to auditory, ventral premotor and inferior frontal regions (Niziolek & Guenther 2013. The Journal of Neuroscience, 33(29), pp. 12090-12098) in both hemispheres (see methods for details).*” (p. 8).

Based on your comment we checked again the labelling of activated brain regions. The triangular activation cluster extended into the frontal operculum but not into the insula. We corrected this in the text. Note that the ROIs have a diameter of 10mm, which justifies labelling activation clusters according to their respective subregions.

To avoid confusion, we now use “ROI” only for defining the literature-based search space. “Spheres” are used to describe voxels whose activity was extracted for correlation and connectivity analyses. In the latter case, we label the identical set of voxels as “seeds”.

3. I could not figure out the relationship between the data shown in Fig 3 and Table 2. Unless I am mistaken they are supposed to show the same analysis (as indicated on line 202), but they don’t seem to correspond. For instance, in Table 2, the left anterior STG shows modulation with three different left-hemisphere areas during the spectral manipulation; so there should be three orange lines emanating from this region in the upper right panel of Fig 3; but there are only two such lines. The left posterior STG is indicated as having an increased connectivity with the left TPJ, also in the spectral condition, so there should be an orange line for that, but it’s not present. Conversely, in the figure there are two orange lines for the right posterior STG, but this latter region does not exist in the table. There are other discrepancies too. More generally I was not very convinced of the value of the interhemispheric connectivity findings. I am not sure what hypothesis this analysis was supposed to test, and the discussion barely mentions these findings. I would question whether it is meaningful to include these findings without a better framing of the intention behind them, or of their interpretation.

Thanks for pointing out this mistake. In the previous Table 2 we had a typo (L instead of R) and one connection was mistakenly not reported in the Table. We corrected Table 2. The old

Figure 3 correctly represented our results. Based on your suggestion, we more clearly attribute connections to their respective seeds and we eliminated the interhemispheric connectivity results. Please note that the connectivity between the right pSTS and the right Heschl gyrus and between the left aSTS and the left inferior parietal sulcus is not visible in the rendering. We report them in the table and in the Figure legend. Note that the seemingly right anterior temporal pole connection actually reflects a part of the right inferior frontal cluster that extends into the orbitofrontal cortex and only appears to be a connection with the anterior temporal pole. Because we could not render the results differently, we clarify this point in the Figure legend. We now write: “*Note that the right inferior frontal cluster extends from the lateral surface into the orbitofrontal cortex which seemingly gives rise to two clusters in the rendering. Spectral adaptation also increased connectivity between L aSTS and L IPS and between R pSTS and HG (Table 2, these deeper structures are not rendered on the surface)*”.

We agree that the STS would be a more correct label than the STG. We corrected this throughout the manuscript.

Minor items:

Although perhaps not essential, it might be nice for the reader to have an illustration of the stimuli used and the nature of the spectral or temporal manipulation applied. This could be done with spectrograms of the stimuli before and after the modification was made for example.

We now provide a supplementary Figure 2 that illustrates the experimental manipulations.

I found this wording (line 225) very odd: “Our findings ask for a specification of prevailing speech production models...” Perhaps the authors mean to say that their findings suggest a modification, or an addition to existing models.

We changed the wording and write: “*Our findings ask for a modification of prevailing speech production models*” (line 215).

“principals” should be “principles” on line 344

Corrected.

Response to reviewer 3

This manuscript investigates the claim that during speaking, auditory feedback control in the spectral domain (e.g. compensating for mismatches in formants, pitch, or spectral centroid) is differentially processed by the right hemisphere, while control in the temporal domain (e.g. compensating for mismatches in phoneme or syllable timing) is differentially processed by the left hemisphere. This is a straightforward claim that makes clear predictions beyond those made by existing models of speech production, which do not delineate separate contributions of temporal or spectral feedback processing circuits. These predictions are tested in two well-designed complementary experiments and are borne out by the authors' data. First, adaptation to altered auditory feedback is differentially enhanced or diminished based on the ear that is receiving the feedback alteration: there is more adaptation when hearing a spectral mismatch in the left ear (right hemisphere) or a temporal mismatch in the right ear (left hemisphere). Second, the authors present here the first neuroimaging study of temporal perturbations to speech feedback, showing that responses are left-lateralized in the ROIs that are responsive to these perturbations, while responses to spectral perturbations are right-lateralized. These are novel results that touch on both sensorimotor control of speaking and hemispheric specialization for auditory stimuli and would be of wide interest to the speech and language community.

The claims would be made more convincing by showing the data, especially the behavioral adaptation, in more detail. In Figure 1, the dichotic listening conditions are collapsed across two groups each (i.e. across vowel and consonant perturbations), and the the binaural condition (black dotted line) seems to be averaging across all four groups (i.e., across both spectral and temporal perturbations). While seeing this aggregate data paints a clear picture of the overall effect, I think it's more appropriate to show spectral and temporal groups separately. These are very different manipulations (an increase in produced F1 vs. a decrease in spectral center of gravity vs. a shortening of phoneme duration) and the raw compensation measurements are on different scales (Hz vs. ms). Even if the compensation evoked by these different types of feedback perturbations is normalized, and is comparable in magnitude once normalized, it's still an overreach to average together these different data types and not show the data from the individual conditions. Since participants either experienced spectral or temporal perturbations, it would make sense to split these data and show all of the four conditions (binaural, left, right, and control) on each plot, at least in a supplemental figure if not in the main text. (Further, in the current Figure 1, the error bars for the five lines are difficult to discern; it would help to offset them horizontally or include error bar "caps" so it is easier to tell where the error bars for each line begin and end.)

We followed all these excellent suggestions that are related to reviewer 1 comment 4 and 6 and reviewer 2 comment 1.

The adaptation data from the fMRI study are not shown at all, so there is no way to evaluate the magnitude of the behavioral effects in this study. In particular, since different statistical reliabilities are reported for the spectral and temporal groups ($p = 0.05$ vs. $p < 0.001$), a behavioral data figure would be useful to evaluate the compensation each type of perturbation evoked. These separate statistics for spectral and temporal perturbations are not given for the behavioral study; it is unclear why

they are separated here and not in the behavioral study, since in both studies there were separate groups of participants for the two types of feedback perturbations.

We added the behavioral data of the fMRI experiment (see also our response to comment 1 of reviewer 1 and 2). We now report statistics using one model that includes all participants which allows testing for differences between compensation for spectral and temporal perturbations, both in the analyses of the behavioral and the fMRI experiment. In the fMRI experiment, the analogous model as in the behavioral experiment revealed compensation for spectral and temporal perturbations although with a close to threshold difference in the degree of compensation between spectral and temporal perturbations. We now write: *“Participants in the fMRI study also changed their speech production upon perturbation relative to baseline (Figure 3a) with marginal smaller changes for spectral compared to temporal perturbations ($F(1, 42)_{type} = 3.61, p = 0.064$; $estimate_{spectral} = 0.0199$ SE = 0.006, $t(42) = 2.11, p = 0.04$; $estimate_{temporal} = 0.027$ SE = 0.006, $t(42) = 4.76, p < 0.001$).”*

We additionally tested acoustic features for condition differences within the perturbation run by contrasting against the control condition. While acoustic features significantly differed between the temporal perturbation and the control condition, carry over effects from the experimental to the control condition rendered differences in production with altered feedback relative to normal feedback in the same run non-significant for the spectral group. Given the results of the aforementioned analyses and the observed strong right-lateralized activations during compensation for spectral perturbations together with previous reports of the same protocol (Tourville et al 2008. NeuroImage, 39 (3), pp. 1429-1443; Niziolek & Guenther 2013. The Journal of Neuroscience, 33(29), pp. 12090-12098) let us be confident that spectral adaptation was in place also during fMRI.

We now write: *“When comparing speaking with altered auditory feedback to the control condition instead of pre-perturbation values, only compensation to temporal perturbations reached significance ($F(1, 42)_{SpectralvsTemporalXNormalvsAltered} = 17.14, p < 0.001$; $estimate_{spectral} = 0.003$ SE = 0.004, $t(42) = 0.761, p = 0.45$; $estimate_{temporal} = 0.027$ SE = 0.004, $t(42) = 6.36, p < 0.001$). This resulted from carry over effects from the experimental to the control condition in the spectral perturbation group (see Figure 3a).”*

Relatedly, since neither behavioral data nor correlation plots are included for the fMRI study, there is no sense of range for the adaptation effects and how they contribute to the observed neural activation. Furthermore, showing a correlation between adaptation and whole-brain activity (perturbation > no perturbation), rather than the correlation between adaptation and cross-hemisphere differences, would be a more primary measure and useful for drawing conclusions about how different regions are recruited in the course of auditory-motor learning.

Reg. behavioral data during fMRI, please see above. We added the correlation plots, as suggested, for correlations between acoustic values and brain activity instead of interhemispheric balance (see also comment 2 of reviewer 2). However, we correlated activity in spheres around condition effects that were also used for the connectivity analyses to focus the analysis on feedback control-related activity.

Methods:

- Throughout the paper, "perturbations" and "manipulations" seem to be used interchangeably. This is somewhat confusing especially given the abstract where they

might be interpreted contrastively (the abstract twice contrasts "temporal manipulations" with "spectral perturbations").

We now use "perturbation" throughout the manuscript.

- Were the 40 participants in the behavioral experiment evenly divided among the four experimental groups (10 per group)?

Yes, they were, even after exclusion of 4 subjects which we failed to mention in the previous version of the manuscript. We added this information on p. 16.

- Line 400: "We have chosen the high vowel /i/..." and line 515: "The vowel /i/ was perturbed..." -- Do you mean the vowel /ɪ/? According to lines 409 and 444, spoken stimuli do not contain the vowel /i/.

Indeed, we meant the lax vowel /ɪ/ and corrected this throughout the manuscript.

- How were the written stimuli spelled when presented to participants?

Stimuli were spelled with a grapheme to phoneme relation that would trigger the respective sound in German, e.g, *sch* [ʃ], *ch* [ç] and *i* [ɪ] (in contrast to "ie", "ih" or "ieh" which would mark the tense vowel [i]).

- The total fMRI trial length was reported to be 8 seconds, but the trial timeline seems to have events that sum to 9 seconds (2s acquisition + 0.5-1.5s pause + 2s CVC presentation + 3.5-4.5s pause).

This was a typo. The pause was jittered between 2.5 and 3.5 seconds. Corrected.

- Was perceived pitch also altered by Audapter during the spectral perturbation of /ʃ/? Did participants compensate for this by lowering their F0?

Our perturbation only targeted the fricative portion, leaving the pitch of the vowel portion unaltered, which we mention here: "*The online status tracking function of Audapter was used to restrict feedback perturbations to either the vowel or the fricative in the syllable*" (p. 22). We additionally checked whether participants lowered their vowel's F0 in response to spectral perturbations of the fricative. This was not the case. We corrected the wording in parts of the methods which could have led to misunderstandings.

Results:

- Any overlap of regions responding to both spectral and temporal perturbations should be denoted in Figure 2a (it is difficult to tell if there is no overlap or if activation from one group of subjects is merely on top of the other).

There is some overlap of activity in auditory regions between the spectral and temporal group. We increased the transparency of overlays in Figure 3 (old Figure 2) and now mention the overlap in the figure legend.

Discussion:

- Lines 281-283: The SFC model doesn't propose a single left-lateralized internal auditory-motor interface in the left TPJ. While Hickok et al. do claim Spt as the

primary sensorimotor integration area, Houde and Nagarajan's model does not commit to this laterality: "Note that although, for simplicity, only the neural substrate in the left hemisphere is shown here, we would expect the full network of the neural substrate to include analogous areas in the right hemisphere as well. At this point, the SFC model is agnostic regarding hemispheric dominance in the proposed neural substrate."

We now specify that the claim regarding a left-lateralized TPJ stems from Gregory Hickok which we cite in this context: "*The observation of, both, a left and a right internal fronto-temporal loop appears in contradiction with the proposal of a single left lateralized internal auditory-motor interface in the left TPJ (Hickok & Houde 2011. Neuron 69 (3), pp. 407-422; Hickok 2012, Journal of Communication Disorders, 45, pp 393-402.)*" (p. 12). We now also specify more clearly that Houde's model did not propose such a left-lateralization: "*Previous theoretical models propose a single auditory feedback controller, either in the right hemisphere (Tourville et al 2011. Language and Cognitive Processes 26, pp 952-981) or did not specify the contributions of the two cerebral hemispheres (Houde & Nagarajan, 2011. Frontiers in Human Neuroscience 5, 82.)*" (p. 10).

- Line 337-338: "increased executive control" is a somewhat speculative interpretation based only on increased coupling between auditory association cortices and fronto-parietal control networks.

We eliminated this analysis, see comment 3 of reviewer 2.

Reviewers' Comments:

Reviewer #1:

Remarks to the Author:

I would like to thank the authors for their thorough revision of the manuscript, especially the clarifications and much improved figures. My concerns have been sufficiently addressed.

Signed: Dr Anne Keitel, University of Dundee, UK

Reviewer #2:

Remarks to the Author:

The revisions have addressed all my concerns and I think, the issues raised by the other reviewers as well. The paper seems much clearer, especially after correction of a few errors and clarifications provided by the authors, so I am happy to recommend acceptance. I have only two small comments:

1. There is a paper just out (Albouy et al, Science, 2020) which is directly relevant to the main point of the study, showing that temporal modulations are decoded more accurately in left auditory cortex and spectral modulations on the right. Seems like it would be worthwhile to mention it as it fits well with the authors' model.
2. I had previously questioned the sentence "Our findings ask for a specification of prevailing speech production models..." which the authors have modified. Except that the part of the sentence I found odd was the use of the word "ask" which the authors have left in. I don't want to be so picky but it still sounds weird to me to say that findings ask for something. Perhaps a better way to express it would be to say that the findings suggest that a modification of current models is needed, or something like that. Anyway, small detail.

Reviewer #3:

Remarks to the Author:

The paper has been substantially improved by the inclusion of more data and clarification of the fMRI analyses, which strengthens the authors' claims of hemispheric specialization for spectral vs. temporal feedback control. While the data are now much more detailed, they bring up a few additional questions. First, I still question the choice to combine both spectral and temporal perturbation responses as a single (normalized) dependent measure in the behavioral analysis, while the fMRI analysis used two separate repeated measure ANOVAs separating the spectral/temporal groups (lines 622-623). Why separate the fMRI data but not the behavioral data, especially given that the raw compensation measurements are on different scales (Hz vs. ms) and these experiments involved different participants? This should be justified especially in light of the decision to use separate LMMs for the binaural and dichotic data and to exclude the control data from any analysis, which makes it harder to compare response magnitudes across these conditions in the same participants.

Relatedly, vowel and consonant data are now shown as individual data points (one per participant), which is good, but given Figure 1b and 1c it looks as if adaptation is mostly driven by the vowel (yellow/cyan), not the consonant (red/blue) data. Pooling the two experiments and averaging makes this unclear. It seems important to establish that the consonant manipulation successfully changed consonant productions; otherwise, the whole effect could be due to vowel responses, which would weaken the generalizability of the claim. The LMM showed "a marginal trend that compensation was

larger for perturbations applied to vowel acoustics compared to consonant acoustics", but was the effect on consonant acoustics significant in its own right? Again, since the behavioral experiment participants who experienced perturbations to consonants were a separate group from those who experienced perturbations to vowels, and the data are independent, it seems important to establish that there was an effect on consonant acoustics, not merely that there was an effect in the pooled data and that the null hypothesis that vowels were different from consonants could not be rejected ($p = 0.08$).

Finally, was the factor of target of the feedback manipulation (vowel/consonant) left out of the list of fixed effects in the fMRI model (line 576)? The following line ("allowing by-subject random slopes for the effect of target of the feedback alteration (vowel/consonant)") implies that it was a factor but I didn't see statistics reported for this factor. The compensation-fMRI correlation analysis is also not described in much detail; were the per-participant compensation values and activity values shown in Supplementary Figure 1 averaged across vowel and consonant conditions?

Methods

- Line 351: What task instructions were not followed for the excluded speaker?

- Line 361: Typo for "Beyerdynamic".

- Lines 392-393: The two lists of words appear to be swapped here (the words in the consonant list should contain /f/ and the words in the vowel list should contain /ɪ/, since these were targeted in these two conditions, respectively).

- Line 632: Typo for "a priori".

Figures

- In Figures 1 and 2 and Figure 3a, the y-axis labels contain "(%)" but are not in percent as presented (the normalized value is 1.0, not 100).

Response to Reviewer

Reviewer 1:

I would like to thank the authors for their thorough revision of the manuscript, especially the clarifications and much improved figures. My concerns have been sufficiently addressed.

Answer: We thank the reviewer for her contribution.

Reviewer 2:

The revisions have addressed all my concerns and I think, the issues raised by the other reviewers as well. The paper seems much clearer, especially after correction of a few errors and clarifications provided by the authors, so I am happy to recommend acceptance.

Answer: We appreciate this recommendation.

I have only two small comments:

1. There is a paper just out (Albouy et al, Science, 2020) which is directly relevant to the main point of the study, showing that temporal modulations are decoded more accurately in left auditory cortex and spectral modulations on the right. Seems like it would be worthwhile to mention it as it fits well with the authors' model.

Answer: Thank you for pointing this out. Including this paper was also our plan after having read the publication.

2. I had previously questioned the sentence “Our findings ask for a specification of prevailing speech production models...” which the authors have modified. Except that the part of the sentence I found odd was the use of the word "ask" which the authors have left in. I don't want to be so picky but it still sounds weird to me to say that findings ask for something. Perhaps a better way to express it would be to say that the findings suggest that a modification of current models is needed, or something like that. Anyway, small detail.

Answer: We revised the sentence. It reads now: “*Our findings suggest a modification of prevailing speech production models*“ (p. 11).

Reviewer 3:

The paper has been substantially improved by the inclusion of more data and clarification of the fMRI analyses, which strengthens the authors' claims of hemispheric specialization for spectral vs. temporal feedback control. While the data are now much more detailed, they bring up a few additional questions. First, I still question the choice to combine both spectral and temporal perturbation responses as a single (normalized) dependent measure in the behavioral analysis, while the fMRI analysis used two

separate repeated measure ANOVAs separating the spectral/temporal groups (lines 622-623). Why separate the fMRI data but not the behavioral data, especially given that the raw compensation measurements are on different scales (Hz vs. ms) and these experiments involved different participants? This should be justified especially in light of the decision to use separate LMMs for the binaural and dichotic data and to exclude the control data from any analysis, which makes it harder to compare response magnitudes across these conditions in the same participants.

Answer: We thank the reviewer for pointing this out. We agree that we could have better explained the choice of two separate LMMs for the binaural and dichotic conditions in the previous version of the manuscript. The first LMM on binaural data served to check whether the spectral and temporal perturbations induced compensatory responses. The second LMM on dichotic data tested whether the produced speech features in response to dichotically presented spectral and temporal auditory feedback perturbations depended on which ear received the perturbed auditory feedback, the central research question in this experiment. Answering this question does not require inclusion of the control condition into the model of dichotic conditions, which was the reason why we previously did not model the control condition in the binaural data model, either, for sake of consistency. We fully understand that this choice is debatable and we now followed the reviewer's suggestion to include the control condition into the binaural data model, because we use this model to demonstrate the efficacy of both the spectral and temporal perturbations. We now justify this choice more clearly (pages 25 and 26) and provide statistics, both for comparisons against pre-perturbation values and against the control condition (pages 6 and 7).

We included data from the spectral and temporal perturbation groups into one model of the binaural conditions to be consistent with the dichotic conditions model which requires inclusion of both groups to test for the ear x perturbation type interaction. The statistical test of this interaction requires normalizing spectral and temporal measures and comparing relative production changes. All normalized data points are visualized and inspection let us be confident that relative spectral and temporal production changes can indeed be compared. We now state more explicitly that raw values differ and that normalized values were required to compare responses to spectral and temporal perturbations: *"This results in comparable values of relative production changes that were used for statistics, while the raw values are in different units (Hz and ms)."* (page 25)

Based on your comment, we additionally checked whether the effects in the binaural model including both data from the spectral and from the temporal perturbation groups were equally observed when data were modelled separately. This was the case (see page 6), suggesting that using one model did not create spurious results based on potentially differently scaled effects in the two groups.

We prefer to be consistent in the way we model the behavioural data in the different experiments and accept that fMRI data were modelled separately for the spectral and temporal perturbation groups. Random effects analyses in the standard imaging analysis tool SPM do not permit inclusion of within- and between-subject factors in a single model (which is required for the statistical analysis of the behavioural experiment), because of the way error terms are calculated. fMRI data models are therefore limited and need to be broken down to models with correct error terms.

Relatedly, vowel and consonant data are now shown as individual data points (one per participant), which is good, but given Figure 1b and 1c it looks as if adaptation is mostly driven by the vowel (yellow/cyan), not the consonant (red/blue) data. Pooling the two experiments and averaging makes this unclear. It seems important to establish that the consonant manipulation successfully changed consonant productions; otherwise, the whole effect could be due to vowel responses, which would weaken the generalizability of the claim. The LMM showed "a marginal trend that compensation was larger for perturbations applied to vowel acoustics compared to consonant acoustics", but was the effect on consonant acoustics significant in its own right? Again, since the behavioral experiment participants who experienced perturbations to consonants were a separate group from those who experienced perturbations to vowels, and the data are independent, it seems important to establish that there was an effect on consonant acoustics, not merely that there was an effect in the pooled data and that the null hypothesis that vowels were different from consonants could not be rejected ($p = 0.08$).

Answer: We wish to emphasize that it was our intention to report observations that do not depend on the phoneme that was manipulated. Based on potential (but disputed) hemispheric differences in the processing of vowels and consonants we wanted to be sure that lateralization effects were not driven by the use of only one manipulated phoneme. In the dichotic conditions, neither the interaction between phoneme and ear ($p = 0.244$) nor the interaction between type, phoneme and ear was significant ($p = 0.14$). The fMRI data showed consistent activity and lateralization related with compensation for consonants and vowels (conjunction analyses), which supports the notion of *lateralized* spectral and temporal speech feedback control for both consonants and vowels.

We do not wish to imply that vowel- or consonant-specific processes do not exist, but they are not in the focus of this study. The marginal trend for an overall larger compensation for vowel compared to consonant perturbations was only observed in the dichotic conditions which suggests that compensation for fricative perturbations were slightly weaker when they were experienced only in one ear. Indeed, the requested post hoc for compensation of dichotic consonant perturbations did not reach significance ($p = 0.31$). Yet, consonant perturbations clearly induced compensation as evidenced in the behavioural data of the fMRI experiment ($p < 0.001$). We value your concern, report these post hocs and write: "*There was a marginal trend that compensation was overall larger for dichotically presented perturbations applied to vowel compared to consonant acoustics ($F(1, 31)_{\text{VowelvsConsonant}} = 3.18, p = 0.08, estimate_{\text{vowel}} = 0.39, SE = 0.012, t(36) = 3.33, p = 0.002; estimate_{\text{fricative}} = 0.013, SE = 0.012, t(36) = 1.03, p = 0.31$). In isolation, this finding could potentially indicate that primary vowel rather than consonant perturbations induced compensatory responses in the dichotic conditions. However, the target phoneme of the perturbation (vowel or consonant) did not significantly influence the lateralization effect ($F(1, 31)_{\text{VowelvsConsonantXLeftvsRight}} = 1.41, p = 0.244, F(1,31)_{\text{VowelvsConsonantXLeftvsRightXSpectralvsTemporal}} = 2.35, p = 0.14$).*" (page 6).

Please note that we do not pool results of different experiments, but report results of a mixed within and between-subject design. The study was designed that way because a pure within subject design would have required 16 conditions with a similar number of contextual cues per subject. So far, no studies have been published that use more than 3-4 contextual cues in parallel. 16 conditions per subject would have increased the duration of the experiment to values that induce fatigue and attentional effects. We now write: "*16 experimental*

manipulations were studied in a mixed within and between subject design to reduce the number of conditions per participant, which is important because parallel implicit learning of new auditory-motor associations has only been reported for up to three different perturbations so far (Rochet-Capellan & Ostry, 2011). Participants were evenly divided across four experimental groups that differed with respect to the acoustic property which was altered throughout the experiment. Participants either experienced spectral or temporal perturbations of the vowel or the consonant in their auditory speech feedback. Each participant experienced four different conditions (binaural unaltered feedback, binaural altered feedback and two dichotic conditions).“ (pages 16 and 17)

Finally, was the factor of target of the feedback manipulation (vowel/consonant) left out of the list of fixed effects in the fMRI model (line 576)? The following line ("allowing by-subject random slopes for the effect of target of the feedback alteration (vowel/consonant)") implies that it was a factor but I didn't see statistics reported for this factor.

Answer: Thanks for pointing this out. We corrected the model of the fMRI behavioural data and now also included the target phoneme as fixed effect into the model. There was no significant effect of phoneme ($F(1, 43)_{\text{VowelvsFricativ}} = 0.007$, $p = 0.93$) and significant compensation for perturbations applied to, both, vowel and fricative acoustics ($\text{estimate}_{\text{vowel}} = 0.02$, $\text{SE} = 0.01$, $t(42) = 1.99$, $p = 0.05$; $\text{estimate}_{\text{fricative}} = 0.021$, $\text{SE} = 0.005$, $t(42) = 4.03$, $p < 0.001$) (page 8).

The corrected single model for behavioural data during fMRI now also includes the control condition (see comment #1). These changes increase the previous p value for compensation of spectral perturbations from 0.04 to 0.09. As mentioned in our previous response to reviewers, the observed strong right-lateralized activation during compensation for spectral perturbations together with previous reports of the same protocol (Tourville et al., 2008; Niziolek and Guenther, 2013) let us be confident that spectral compensation was also in place during fMRI. We report this effect as marginal trend and are convinced that the similarity of the associated fMRI activation patters with previous studies documents that the change is not critical to the relevance of our findings.

The compensation-fMRI correlation analysis is also not described in much detail; were the per-participant compensation values and activity values shown in Supplementary Figure 1 averaged across vowel and consonant conditions?

Indeed, compensation and activity values were averaged across vowel and consonant conditions. We now write: *“The relationship between participants’ individual degree of compensation to spectral and temporal feedback perturbations and individual activity in spectral and temporal feedback control areas was assessed using Pearson’s correlations. In analogy to the aforementioned fMRI analyses we did not dissociate vowel and consonant effects and correlated averaged vowel and consonant productions with averaged activity during vowel and consonant perturbations.”* (page 30) and also mention this approach in the results section more explicitly (page 9).

Methods

Line 351: What task instructions were not followed for the excluded speaker?

The participant did not follow instructions to read the displayed words. Instead, he intonated them.

Line 361: Typo for "Beyerdynamic"

Corrected.

Lines 392-393: The two lists of words appear to be swapped here (the words in the consonant list should contain /j/ and the words in the vowel list should contain /i/, since these were targeted in these two conditions, respectively).

You are right. Thank you.

Line 632: Typo for "a priori"

Corrected.

Figures

In Figures 1 and 2 and Figure 3a, the y-axis labels contain "(%)" but are not in percent as presented (the normalized value is 1.0, not 100).

Corrected.

Reviewers' Comments:

Reviewer #3:

Remarks to the Author:

The authors have conscientiously responded to all my concerns in the current revision and I am happy to recommend the paper.